# The translational landscape of ground state pluripotency

Yaser Atlasi[1,2,7,8 ✉], Seyed Mehdi Jafarnejad[2,7], Christos G. Gkogkas[3], Michiel Vermeulen[4], Nahum Sonenberg[5,8 ✉] & Hendrik G. Stunnenberg [1,6,8 ✉]

Translational control plays a central role in regulation of gene expression and can lead to significant divergence between mRNA- and protein-abundance. Here, we used genome-wide approaches combined with time-course analysis to measure the mRNA-abundance, mRNA-translation rate and protein expression during the transition of naïve-to-primed mouse embryonic stem cells (ESCs). We find that the ground state ESCs cultured with GSK3-, MEK-inhibitors and LIF (2iL) display higher ribosome density on a selective set of mRNAs. This set of mRNAs undergo strong translational buffering to maintain stable protein expression levels in 2iL-ESCs. Importantly, we show that the global alteration of cellular proteome during the transition of naïve-to-primed pluripotency is largely accompanied by transcriptional rewiring. Thus, we provide a comprehensive and detailed overview of the global changes in gene expression in different states of ESCs and dissect the relative contributions of mRNA-transcription, translation and regulation of protein stability in controlling protein abundance.

[1] Department of Molecular Biology, Faculty of Science, Radboud University, Radboud Institute for Molecular Life Sciences (RIMLS), 6525 GA Nijmegen, The Netherlands. [2] Patrick G. Johnston Centre for Cancer Research, Queen's University Belfast, Belfast, UK. [3] Centre for Discovery Brain Sciences, and The Patrick Wild Centre, The University of Edinburgh, Edinburgh, UK. [4] Department of Molecular Biology, Faculty of Science, Radboud Institute for Molecular Life Sciences, Oncode Institute, Radboud University Nijmegen, Nijmegen, The Netherlands. [5] Department of Biochemistry and Goodman Cancer Research Centre, McGill University, Montreal, QC H3A 1A3, Canada. [6] Present address: Princess Maxima Centre for Pediatric Oncology, Utrecht, The Netherlands. [7] These authors contributed equally: Yaser Atlasi, Seyed Mehdi Jafarnejad. [8] These authors jointly supervised this work: Yaser Atlasi, Nahum Sonenberg, Hendrik G. Stunnenberg. ✉email: yaser.atlasi@ncmls.ru.nl; nahum.sonenberg@mcgill.ca; h.stunnenberg@ncmls.ru.nl

Although the central dogma of molecular biology (DNA ↔ RNA → protein) has been described decades ago, the relative contributions of major layers of post-transcriptional gene regulation (i.e. mRNA turnover, translational, and post-translational control) in many biological processes remain largely unknown. Several lines of evidence demonstrated that post-transcriptional gene regulation could result in significant divergence between mRNA and protein abundances[1–5]. For instance, quantitative measurements of RNA and protein levels revealed that for ~60% of genes, changes in mRNA abundance cannot accurately predict the variations in protein abundance in mammalian cells[6]. Consistent with this notion, parallel measurements of mRNA expression and protein levels as well as mRNA and protein turnover suggest the major impact of translational control in regulating the proteome landscape[6–8]. In this regard, translational control was postulated to enable prompt response to internal and external stimuli before the less rapid transcriptional reprogramming could take place[8–10].

Translational regulation plays a particularly important role during early embryonic development; during the time window of near complete shutdown of transcription after fertilization, mRNA translation serves as the main layer of gene regulation by which the maternally stored mRNAs are translated in a highly spatiotemporally-controlled manner[11]. Furthermore, maintenance of the undifferentiated status in both adult and embryonic stem cells (ESCs) is associated with low mRNA translation rates and restricted protein synthesis[12–15]. Accordingly, the process of differentiation coincides with a marked increase in general mRNA translation rate. For example, differentiation of ESCs to embryoid bodies results in an increased [35S]methionine incorporation by ~2-fold and polysome density by ~60%[12]. In addition to the global changes, selective translational control of specific mRNAs also plays a critical role in ESCs. For instance, tight translational control of the Yy2 mRNA is critical for maintenance of the mouse ESC self-renewal[16]. Similarly, translational repression of the mitogen-activated protein kinase kinase kinase 3 (Map3k3) and son of sevenless homolog 1 (Sos1) is required for blocking the differentiation of mouse ESCs[17].

ESCs, therefore, rely on tight regulation of mRNA translation in order to maintain their self-renewal and pluripotency. The current knowledge on the translational control in mouse ESCs, however, is mainly based on conventional ESCs cultured in serum/LIF (SL) medium[10,12,16,17]. These undifferentiated cells are metastable and show features of both pre-implantation and early post-implantation epiblast (reviewed in ref. [18]). In addition to SL, ESCs can also be maintained in serum-free medium supplemented with LIF and two small-molecule inhibitors (2iL) which include the GSK3 inhibitor CHIR99021 (CH) and the MEK-inhibitor PD0325901 (PD)[19,20]. 2iL ESCs display major differences with SL cells that concern, among others, different transcriptional profiles, DNA methylation patterns, cell proliferation, and homogeneous expression of pluripotency genes[18,20–23]. Accordingly, 2iL ESCs are less committed cells, better resembling the pre-implantation epiblast, and are often referred to as the ground state pluripotent cells[19]. In contrast to 2iL and SL ESCs, pluripotent cells can be also derived from post-implantation epiblast (e.g. Epiblast stem cells; EpiSCs, or Epiblast like stem cells; EpiLSCs hereafter referred to as EPI). These cells are lineage-primed and strongly downregulate the pre-implantation transcription factors (TFs), show high expression of post-implantation markers, and do not contribute to chimera upon blastocyst complementation (reviewed in ref. [24]).

The transition between different states of pluripotency provides a versatile model for the study of stem cell biology and is of great importance to elucidate early embryonic development. Previous studies investigated the gene expression program at transcriptional and post-transcriptional levels in ESCs and early mammalian development[10,25–28]. These observations indicate the importance of machineries that control mRNA translation and decay in ESCs. These studies, however, mainly focused on single aspects of regulation of mRNA stability and/or translational control (e.g. microRNA-induced repression) in conventionally maintained SL-ESC. The concomitant regulation of transcriptome and translatome, and its global impact on the cellular proteome remains less understood in ground state 2iL-ESCs. Here we employed several genome-wide approaches combined with time-course analysis to measure a detailed and dynamic view of the mRNA abundance, mRNA translation, and protein expression during the transition between different states of ESCs. This multifaceted analysis provides insights into mechanisms controlling the ground-state pluripotency and sheds light on the relative contributions of transcription, mRNA translation, and protein stability in controlling the final protein abundance.

## Results

**Measuring mRNA translation in different states of ESCs.** We sought to study the potential differences in mRNA translation between the ground state 2iL and the more committed SL and EPI states (Fig. 1a). We employed polysome profiling (fractionation of mRNAs based on the number of translating ribosomes, using sucrose-density gradients) and found a considerable increase in the association of cellular mRNAs with polysomes in 2iL when compared to SL and this increase was also observed, although to a lesser extent, when compared to EPI (1.5- and 1.2-fold increase in 2iL when compared to SL and EPI, respectively, Fig. 1b). Given the low translation rate previously reported for the undifferentiated ESCs[12] and the consensus that 2iL represents a less committed state when compared to SL and EPI, the increased polysome density in 2iL was unexpected. We therefore sought to identify the mRNAs that undergo differential translation in different states of pluripotency.

We measured the genome-wide mRNA translation rate by ribosome profiling[29] and in parallel the changes in mRNA expression by RNA-seq in 2iL, SL, and EPI. Ribosome profiling is a next-generation sequencing (NGS)-based method that accurately measures the abundance of ribosome footprints (RFPs; mRNA fragments protected from RNase by the translating ribosomes). The abundance of RFPs for each mRNA serves as a sensitive and quantitative surrogate of its translation efficiency (TE; normalized RFP counts/normalized mRNA counts) and represents a genome-wide measurement of the translation landscape. This technique also enables the acquisition of positional information with nucleotide precision regarding the association of ribosomes with a particular transcript. We collected biological duplicates per cell condition, while parallel cell lysates were used for ribosome profiling or for rRNA-depleted total RNA-seq library preparation. We verified the high quality of generated data using several parameters: (a) the data were highly reproducible among the biological replicates ($r > 0.93$ for RFP and $r > 0.95$ for RNA, Supplementary Fig. 1a, b, Supplementary Data 1), (b) the RFPs define the known coding sequences (CDS) with 12-nt offsets upstream of the translation start codons (which reflects the known distance from the 5′-end of RFPs to the known P-site codons), (c) a high coverage of RFP reads within coding sequence but low coverage at untranslated regions (UTRs), which is specific for RFP but not RNA-seq libraries (Supplementary Fig 1c–e). In total, we detected high-confidence RFPs in 6487 mRNAs in 2iL, SL, or EPI (with library size normalized RFP reads >25 and RNA-seq reads >50 in both replicates). Furthermore, comparing the gene expression profile of 2iL, SL, and EPI ESCs

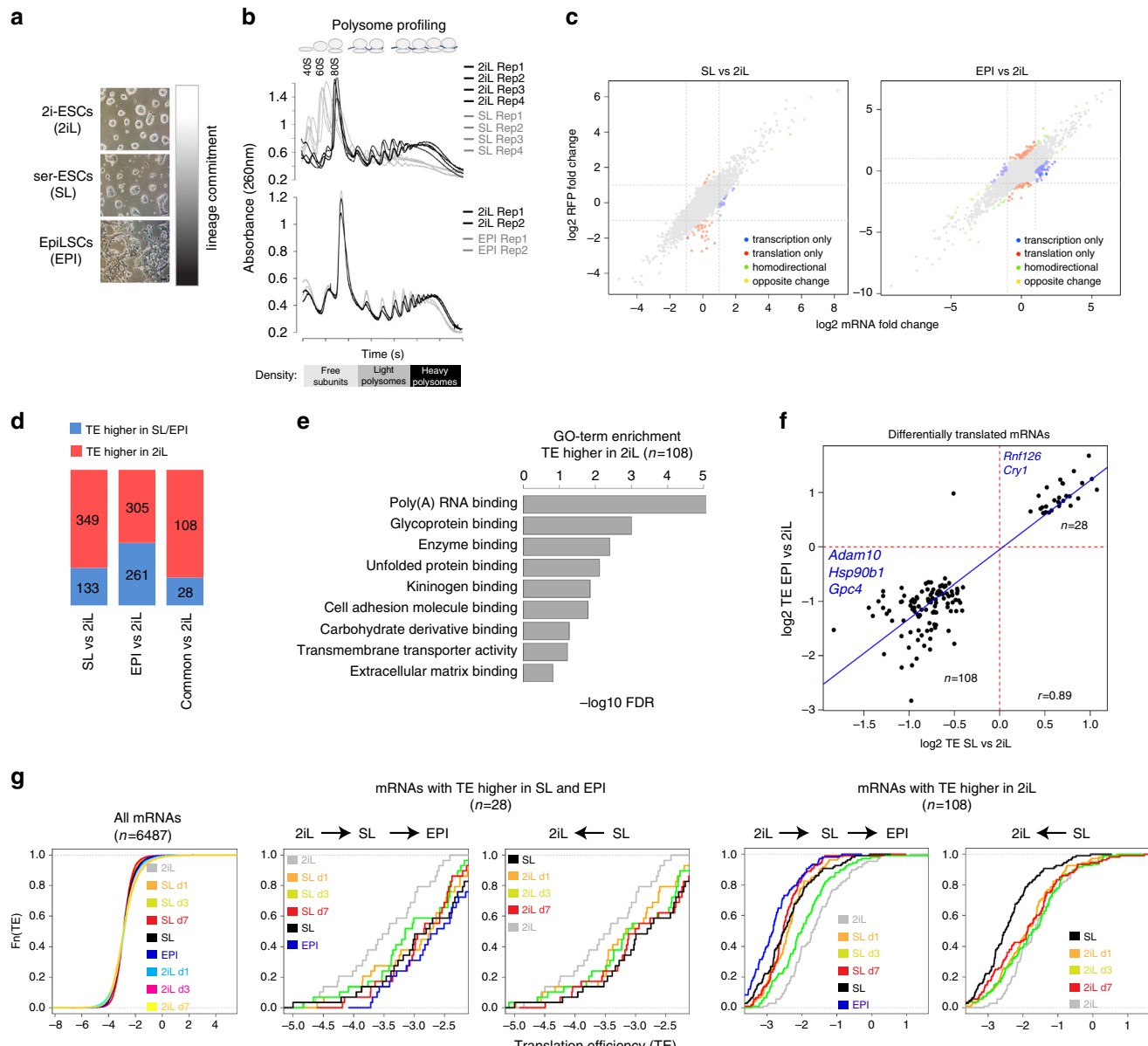

**Fig. 1 The translational landscape of different states of pluripotency. a** Schematic representation of the different ES states employed in this study. Scale bar, 100 μm. **b** Polysome fractionation performed in 2iL, SL, and EPI. $n = 4$ biological replicates for 2iL vs. SL and $n = 2$ biological replicates for 2iL vs. EPI comparison. **c** Scatter plots showing the association between mRNA and RFP fold-change in SL or EPI when compared to 2iL state. Differentially translated mRNAs (FDR < 0.1 and fold-change >1.5 using Xtail analysis pipeline) are highlighted in different colors based on the direction of mRNA and RFP changes. Homodirectional=mRNA and RFP display change in the same direction (up or downregulated). Opposite change=mRNA and RFP display change in opposite directions. $P$ values were adjusted for multiple testing using Benjamin–Hochberg correction, $n = 6487$ mRNAs. **d** Bar plots showing the number of differentially translated mRNAs in SL, EPI, or in both ("common" genes) when compared to 2iL state. Differentially translated mRNAs are defined by FDR < 0.1 and fold-change >1.5 using Xtail analysis pipeline. $P$ values were adjusted for multiple testing using Benjamin–Hochberg correction, $N$ numbers represent mRNAs and are stated in the figure. **e** GO-term enrichment analysis for the "common" genes with higher TE in 2iL cells. Two-sided values were adjusted for multiple testing using Benjamin–Hochberg correction, $n = 108$ mRNAs. **f** Scatter plot showing the correlation of TEs in SL and EPI states. The set of "common" genes were used in this analysis and representative examples were highlighted in the figure. $R$-values represent Pearson correlation coefficients, $N$ numbers represent mRNAs and are stated in the figure. **g** Plots showing the change of TE at different time points of SL-to-2iL or 2iL-to-SL transition. The set of "common" genes were used in this analysis. "All" genes were also included for comparison. $N$ numbers represent mRNAs and are stated in the figure.

and the early developing mouse embryo[30] indicated that 2iL-specifc genes are mainly enriched in pre-implantation embryo whereas the EPI-specific genes are mainly enriched in post-implantation epiblast. SL-specific genes were enriched in both pre- and post-implantation stage embryo reflecting the "meta-stable" state of SL-ESCs with features resembling both naive and primed state pluripotency[18,24] (Supplementary Fig. 2).

We next computed the TE in different conditions and identified a set of mRNAs that show differential TE in SL or EPI when compared to 2iL (FDR < 0.1, fold-change > 1.5, $n = 912$ genes, Fig. 1c). We grouped the mRNAs that showed differential TE in SL (SL genes; $n = 482$), EPI (EPI genes; $n = 566$), and both SL and EPI (common genes; $n = 136$) when compared to 2iL (Fig. 1d). In all the three groups, the majority of genes showed

increased TE in 2iL state. These results are in line with the polysome profiling results and point to a general increase in TE in 2iL when compared to SL and EPI. Gene ontology (GO) analysis revealed that SL genes were mainly enriched in polyA-RNA-binding proteins and included a large number of genes encoding ribosomal proteins ($n = 10$ $Rpl$ genes and $n = 8$ $Rps$ genes, FDR < 0.1, fold change >1.5) (Supplementary Fig. 3a, b) whereas EPI genes and common genes were mainly enriched in polyA-RNA-binding proteins (Fig. 1e, Supplementary Fig. 3a, Supplementary Data 2). In contrast to SL culture, both 2iL and EPI cultures are based on serum-free medium supplemented with small-molecule inhibitors (for 2iL culture) or growth factors (FGF2 and ACTIVIN for EPI culture). Both SL and EPI conditions, however, display increased lineage priming when compared to 2iL state. Thus the "common" set of genes are likely to represent developmentally regulated genes rather than the effect of culture media; we therefore focused on the common gene set for the rest of this study. In almost all the common genes, the change in TE followed a similar direction in both SL and EPI states (cor=0.89) and the vast majority of these genes (108/136, 79%) showed increased TE in 2iL state (Fig. 1f).

The two states of SL and 2iL are interconvertible and can be induced by switching the culture condition from SL-supplemented to 2iL-supplemented medium. We therefore asked whether the change in TE takes place early during SL–2iL transition or whether this change is observed only at the steady-state SL and 2iL. To address this, we performed ribosome profiling at different time points of SL-to 2iL and 2iL-to-SL transitions (Day 1, Day 3, and Day 7 of transition). We found that change in TE is observed as early as Day 1 of SL-to-2iL or 2iL-to-SL transitions, indicating an early response in the form of differential TE during the conversion of the two ES-states (Fig. 1g).

To further validate the results of the ribosome profiling, we measured the changes in translation rate of a number of candidate genes with differential TE in SL and EPI using polysome fractionation followed by quantitative reverse transcription polymerase chain reaction (qRT-PCR), using specific primers against mRNAs of interest. We also examined $Oct4$, which showed no differential TE in our ribosome profiling data, as control. In line with the ribosome profiling results, genes that showed higher TE in 2iL displayed significantly higher association with heavier polysome fractions when compared to SL and EPI (Supplementary Fig. 3c).

Thus, the 2iL ground state is associated with increased ribosome occupancy for a subset of mRNAs, which display reduced TE during early commitment to SL and EPI states.

**Pervasive translational buffering in ground state ESCs**. In order to assess the impact of changes in TE on alterations in protein levels, we measured global protein abundance in SL, 2iL, and EPI and during the SL–2iL transition using mass spectrometry-based proteomics. Label-free quantification[31] identified 4743 protein groups that give rise to 5969 individual proteins across different samples (Supplementary Data 3). Based on this quantitative information, we sought to identify specific patterns of changes in the mRNA expression and TE that would explain the variations in protein abundance among SL, EPI, and 2iL. We integrated the mRNA, RFP, and protein expression and selected a group of proteins that could be accurately quantified in (a) all three 2iL, SL, and EPI states and (b) could be uniquely assigned to their corresponding mRNAs, yielding a total of 3294 uniquely assigned protein-RFP-mRNA groups (Fig. 2a, Supplementary Fig 4a, Supplementary Data 4). We noted that generally, changes in mRNA expression highly correlated with changes in ribosome occupancy

in both SL and EPI when compared to 2iL (Pearson's correlation coefficient=82% and 81%, respectively, Fig. 2b). In contrast, changes in mRNA or RFP expression correlated to a lesser degree with changes in protein abundance (Pearson's correlation coefficient=72% in SL vs. 2iL and 61% in EPI vs. 2iL, Fig. 2b).

At the global level, we observed that in general a change in TEs between different ES states does not lead to concomitant change in protein expression (cor=0.01 in SL vs. 2iL and cor=0.04 in EPI vs. 2iL, Fig. 2c). Strikingly, the vast majority of genes with differential TE showed no change in protein abundance among the different ES-states (75% of "common" genes, 81% of SL genes and 88% of EPI genes, protein fold change <2). Importantly, for these genes we observed that change in TE often takes place in an opposite direction to that of mRNA expression, culminating in similar abundance of the corresponding proteins (Fig. 2d). Thus, the majority of mRNAs with differential TE undergo "translational buffering" in different states of ESCs; a large number of these genes show lower mRNA expression and increased ribosome occupancy in the 2iL state, leading to a comparable net protein level when 2iL is compared to SL or EPI states. Notably, the buffering effect of translational control was not universal as among the mRNAs that display differential TE, we also detected a number of mRNAs such as $Rnf126$, $Trim27$, and $Med21$, for which change in the TE was mirrored in concomitant change in protein abundance (Fig. 2e, Supplementary Fig 4b). To validate these observations, we selected $Rnf126$, which showed the highest change in protein expression (based on mass spectrometry data) in SL and EPI when compared to 2iL, as a representative mRNA. Polysome fractionation followed by qRT-PCR analysis confirmed the shift of mRNA from light to heavy polysome fractions in SL and EPI (Fig. 2f). Consistently, western blot analysis confirmed the increased RNF126 protein abundance in SL and EPI when compared to 2iL ESCs (Fig. 2g).

To probe the mechanism underlying the translational buffering, we analyzed the common features of mRNAs that show similar TE-changes in different culture conditions. There was no significant difference in the length of the 5′- or the 3′-UTRs between the differentially translated mRNAs and a set of random genes that were selected for comparison (Supplementary Fig. 4c). Furthermore, no significant difference in %GC of 5′ UTR regions was found in differentially translated genes when compared to the control set of genes (Supplementary Fig. 4c). However, we found a significantly higher %AU in 3′ UTRs within genes that show higher TE in 2iL ($P$ value=7.28e-05, Supplementary Fig. 4c). Accordingly, a notable enrichment for AU-rich elements (AUUUAU, 59% of genes, $P = 2.9$e-5) was found in this set of translationally upregulated genes. Motif analysis for consensus-binding sites of RNA-binding proteins (RBPs) revealed that the 3′ UTR regions of these mRNAs are strongly enriched, among others, with ELAVL1/2/3/4 binding motifs ($P$ value = 1.1e-06), a protein family known to specifically bind to the AU-rich elements[32] (Supplementary Fig. 4d). In line with this observation, we found significant (FDR < 0.05) downregulation of ELAVL2/4 protein in both SL and EPI when compared to 2iL state (2-fold and 1.7-fold, respectively). These results point to a potential regulatory mechanism that involves the AU-rich elements and their RBPs at the 3′-UTRs of mRNAs that undergo specific translational repression in SL and EPI.

Taken together, the transition to/from ground state 2iL displays translational buffering for a set of mRNAs which ensures the maintenance of stable protein levels. Thus, state-specific translational control contributes minimally to the protein variation during the naive to primed state transition. Instead, this mode of gene regulation may provide a potential mechanism for counteracting the transcriptional noise in response to environmental changes.

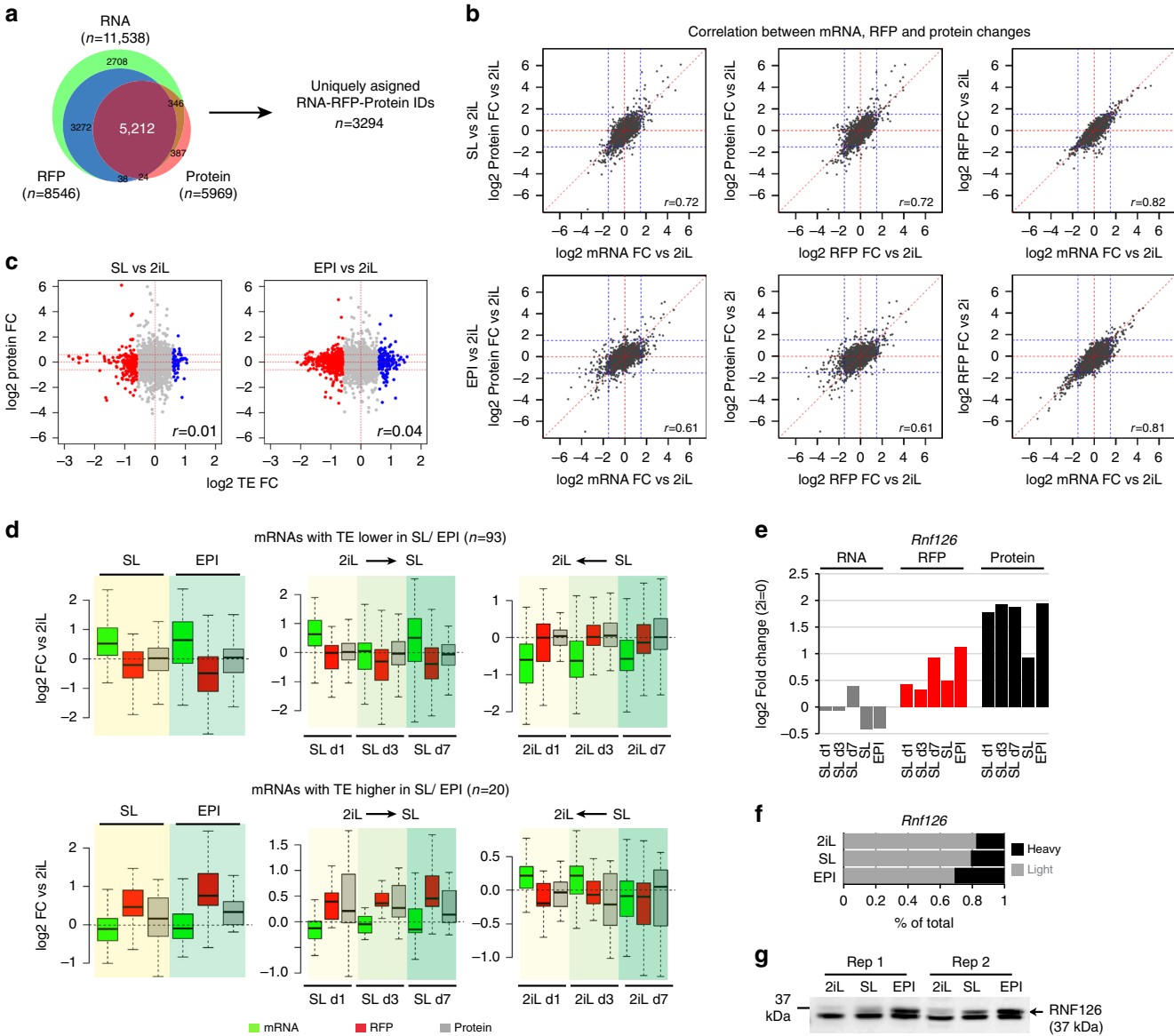

**Fig. 2 Translational buffering maintains a largely stable protein expression in 2iL ESCs. a** Venn diagram showing the overlap between RNAs, RFPs, and proteins detected in SL, 2iL, or EPI states. RNAs with minimum of 50 reads and RFP with minimum of 25 reads in at least one of the ES conditions were selected and intersected with the 5969 detected proteins. Only uniquely annotated proteins that could be detected in all three conditions (to allow for fold change comparison) were maintained, giving rise to $n = 3294$ uniquely assigned RFP–RNA–protein IDs. **b** Scatter plot showing the correlation between fold change in RFP, RNA, and proteins when SL or EPI are compared to 2iL state. Values represent the mean of two highly similar biological replicates. $n = 3924$ RFP–RNA–protein IDs. $R$-values represent Pearson correlation coefficients. **c** Scatter plot showing the correlation between changes in TE and protein expression when SL or EPI are compared to 2iL state. Values represent the mean of two highly similar biological replicates. $n = 3924$ RFP–RNA–protein IDs. $r$-values represent Pearson correlation coefficients. **d** Box plots showing the opposite changes in RNAs and RFPs abundances that result in maintenance of constant protein levels in different states of ESCs and during the SL-to-2iL or 2iL-to-SL transition. Box=25–75th percentile; bar=median; whiskers=5–95th percentile. **e** Bar plot showing the fold change in RNA-, RFP-, and protein-levels of *Rnf126* during 2iL-to-SL transition and in EPI state. **f** Bar plot showing the distribution of *Rnf126* mRNAs in low-density and high-density polysome fractions in different states of ESCs. The qRT-PCR values represent the mean of two highly similar biological replicates. **g** Western blot analysis of RNF126 in different ESC states. Two biological replicates were used per condition.

**Proteome changes are accompanied by transcriptional rewiring.** Since variation in protein levels is the key determinant of gene expression and as the vast majority of changes in protein expression could not be explained by translational control, we next sought to further characterize the differential protein expression during naive to primed ESC transition. The quantitative information obtained at different layers of gene expression and at different time points of naive to primed ESC transition provided a unique opportunity to investigate how the proteome rewiring takes place in different states of pluripotency.

Principal component analysis confirmed the largely similar trajectories of mRNA and protein dynamics during the 2iL to SL and EPI transition (Fig. 3a). Among the 4743 identified protein groups, 26% ($n = 1250$ proteins) showed differential expression in at least one time point of the 2iL to SL to EPI transition (fold change $\geq 3$ and FDR $< 0.05$). When the steady states were compared, 7% ($n = 338$) were differentially expressed between SL and 2iL and 6% ($n = 293$) between EPI and 2iL (Fig. 3a, b). Among the proteins that are differentially expressed in 2iL vs. SL or EPI ($n = 465$), 216 proteins were exclusive to certain ESC state

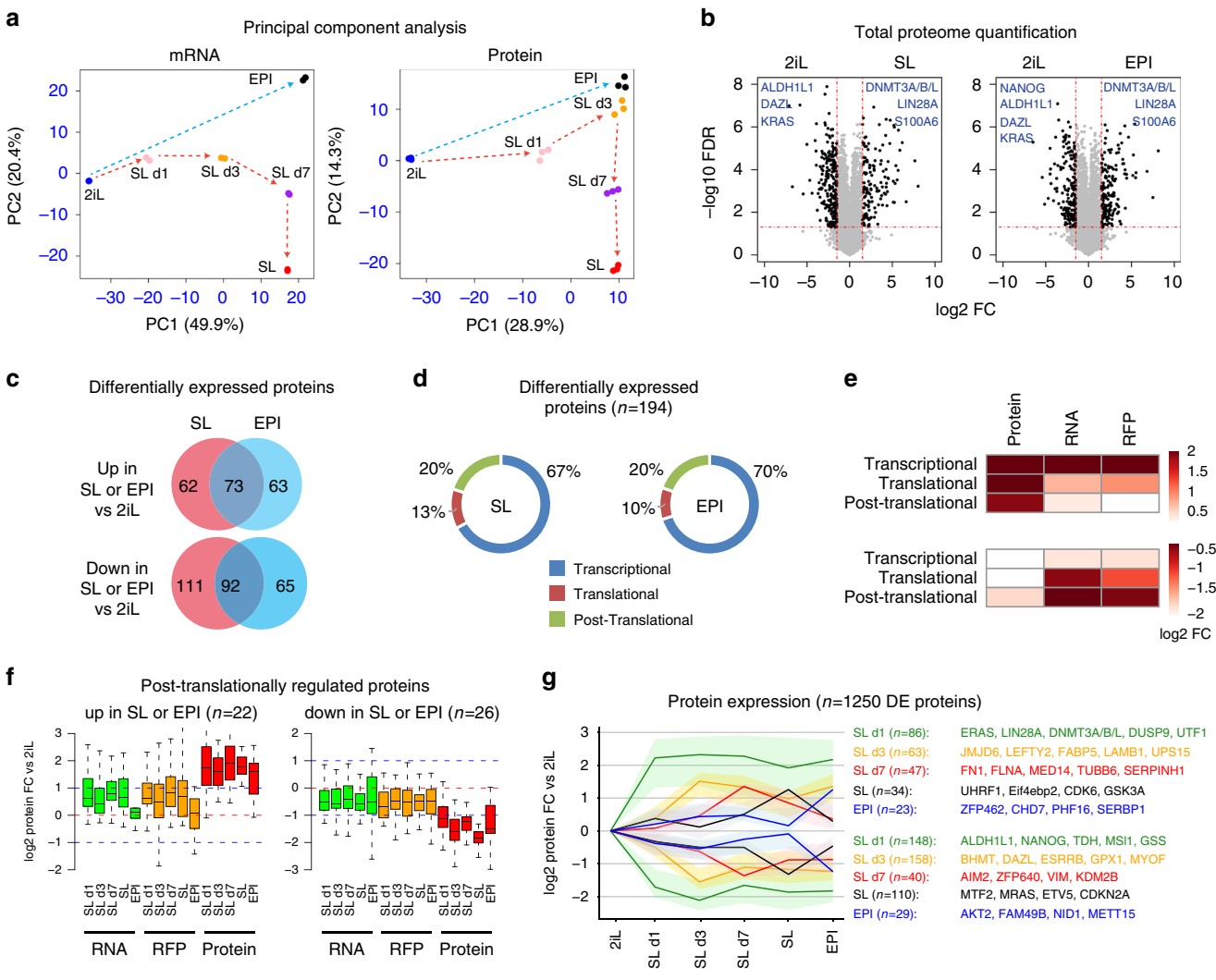

**Fig. 3 Widespread proteome alteration is largely associated with transcriptional rewiring. a** Principal component analysis of the RNA-seq and proteome data from ESCs collected at different time points of the 2iL to SL and EPI transition. $n = 2$ biological replicates per time point for RNA-seq and $n = 3$ biological replicates per time point for total proteome, each dot represents one sample. **b** Volcano plots showing the label-free total protein quantification in 2iL, SL, and EPI conditions. Values represent the mean of three biological replicates. Representative examples are shown. Differentially expressed proteins (FDR < 0.05 and FC ≥ 3) are highlighted in black. Differential expression was calculated using the two-sided $t$-test. **c** Venn diagram showing the overlap of differentially expressed proteins in SL and EPI when compared to 2iL ESCs. $N$ numbers represent proteins and are stated in the figure. **d** Graphs showing the percentage of differentially expressed proteins with or without change in RNA/RFP when SL or EPI are compared to 2iL state. Transcriptional: protein change >3-fold and RNA change >2-fold; translational: RFP change >2-fold and RNA change <2-fold; post-translational: RNA and RFP change <2-fold. **e** Heat-map showing the change in RFPs, RNA, and protein in different categories defined in panel **c**. $n = 194$ differentially expressed proteins. Values represent mean of log2 fold change. **f** Box plots showing the changes in RNA, RFP, and protein levels for the post-translationally regulated genes identified in panel **c** and during the 2iL-to-SL and EPI transition. $n$ represents the number of proteins as indicated in the figure. Box = 25–75th percentile; bar = median; whiskers = 5–95th percentile. **g** Expression pattern of differentially expressed proteins in at least one time point of 2iL to SI and EPI transition. Lines in the plots represent average fold-change of protein expression and shaded bands represent upper 75% and lower 25% quantiles.

and could not be detected in all three ESC states (for example, DAZL and BHMT that are exclusive for 2iL; DNMT3L and TGLN that are exclusive for SL and EPI). RNA-seq analysis showed that 68% of these genes are transcriptionally regulated and showed differential mRNA abundance (fold change >2, FDR < 0.05).

Importantly, we detected 194 differentially expressed proteins in all the three ESC states at the RNA, RFP, and protein levels. For this set of genes, we therefore estimated the contribution of different layers of gene expression in determining the final protein variation. By comparing mRNA, RFP, and protein changes in SL and EPI vs. 2iL, we found that for ~70% of these differentially expressed proteins the change in protein abundance

could be explained by changes in mRNA expression (RNA change >2-fold). Translational control contributed to ~10% of the variation in protein expression for differentially expressed proteins (RFP change >2-fold and RNA change <2-fold). In addition, for ~20% of the differentially expressed proteins, we found that change in protein abundance takes place without evident change in mRNA or RFP expression (RNA and RFP change <2-fold, Fig. 3c–e and Supplementary Fig. 4e). In spite of the differential protein expression, the stable mRNA and RFP abundance in this set of genes suggest a specific post-translational regulation in different states of pluripotency (Fig. 3f). In this group, we detected proteins such as KRAS and MSI1 that are specifically upregulated in 2iL and proteins such as UHRF1 that is

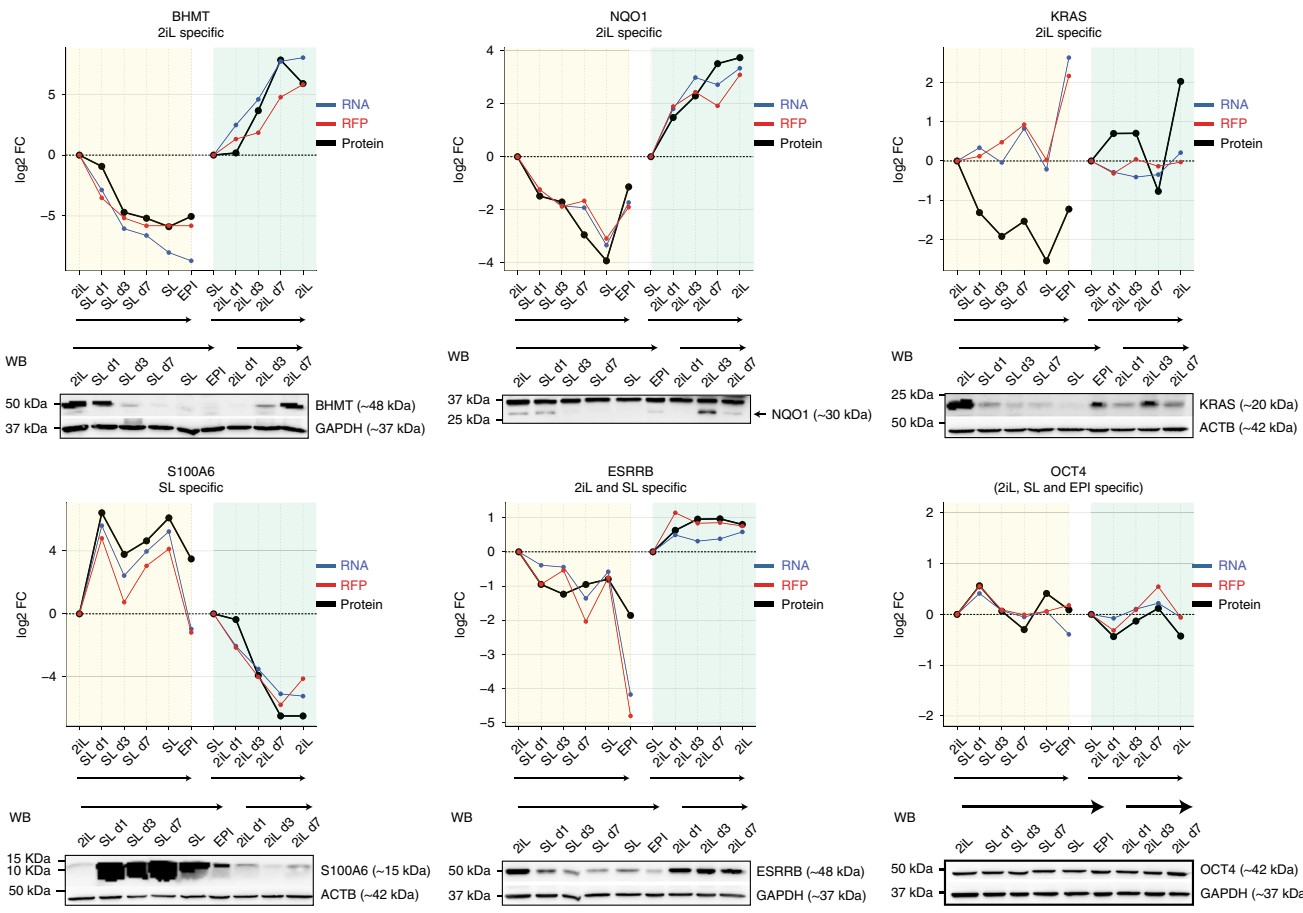

**Fig. 4 Western blot validation of proteome data.** Graphs showing the RNA, RFP, and protein expression levels of selected candidate genes based on RNA-seq, RFP-seq, and LC-MS proteomic analysis. Western blot analysis confirms the proteome data for the selected proteins in 2iL, SL, and EPI states and during 2iL–Sl transition. Candidate proteins included 2iL-specific proteins, BHMT, NQO1, and KRAS; SL-specific protein, S100A6; SL and 2iL-specific protein, ESRRB and a general pluripotency protein (SL, 2iL, and EPI), OCT4. Note that *Kras* is regulated at the post-translational level.

specifically upregulated in SL and EPI states. Differentially expressed proteins between SL and 2iL followed a cascade of gene expression with gradual and continuous change at different time points (Day 1, Day 3, and Day 7) of SL–2iL transition (Fig. 3g). To validate the prteome data we used western blot analysis and examined a set of candidate proteins in 2iL, SL, and EPI states, and during 2iL–SL transition. These proteins included: 2iL-specific proteins, BHMT, NQO1, and KRAS; SL-specific protein, S100A6; SL and 2iL-ESC-specific protein, ESRRB and a general pluripotency protein (SL, 2iL, and EPI), OCT4. Data obtained from western blot analysis largely confirmed the proteomics data for all the examined proteins (Fig. 4).

Thus, the majority of changes in the proteome is accomplished by transcriptional rewiring during naive to primed ESC transition. A set of genes, however, undergo specific regulation at the translational or post-translational levels.

**2iL-specific proteins are enriched for metabolic pathways**. We next sought to functionally annotate the differentially expressed proteins during the naive to primed state transition. The protein signature of the 2iL state was mainly enriched in metabolic pathways (such as glutathione and lipid metabolism) and included proteins such as GSS, IDH1, ACSL1, and DAZL whereas the specific proteins for SL or EPI were mainly enriched in cell adhesion proteins or purine nucleotide metabolism and included proteins such as ITGA3, FLNA, EPCAM, and LIN28A (Fig. 5a, b). We also asked whether the differential proteins in SL or EPI

are enriched for specific protein complexes. By examining the recently annotated compendium of 275 large protein complexes (>5 members[33]) we found that the differentially expressed proteins are highly enriched for the AuroaB-INCENP protein complex, that is involved in regulating the mitotic progression, and the Polycomb protein complex 2 (PRC2) that is involved in H3K27me3 deposition (Fig. 5c, d). For AuroaB–INCENP complex we detected four of the five annotated proteins in which three members (AURKB, AURKC, and MAGED1) were differentially expressed between SL and EPI compared with 2iL. Similarly, we detected 8 of the 11 described core members of the PRC2 complex, in which 4 members (EZH1, SUZ12, MTF2, and AEBP2) were significantly downregulated in SL and EPI states (Fig. 5d). These findings are in line and extend on the previous observations that reported a strong difference in global H3K27me3 and cell cycle progression in different states of ESCs[20,34,35].

Finally, given that the 2iL state is promoted by inhibition of the signaling pathways downstream of the GSK and MEK, we sought to explore the link between the observed differential protein expressions to the upstream signaling pathways. We used CH or PD withdrawal from 2iL to maintain cells for 1 and 3 days with a single inhibitor and generated mass spectrometry-based proteome profiles from the cultured ESCs (Fig. 5e). These cultures represent the early response to the single inhibitor withdrawal and likely reflect direct effects of the small molecule inhibitors. For the majority of proteins that significantly change expression between 2iL and SL/EPI steady states (529/642 proteins),

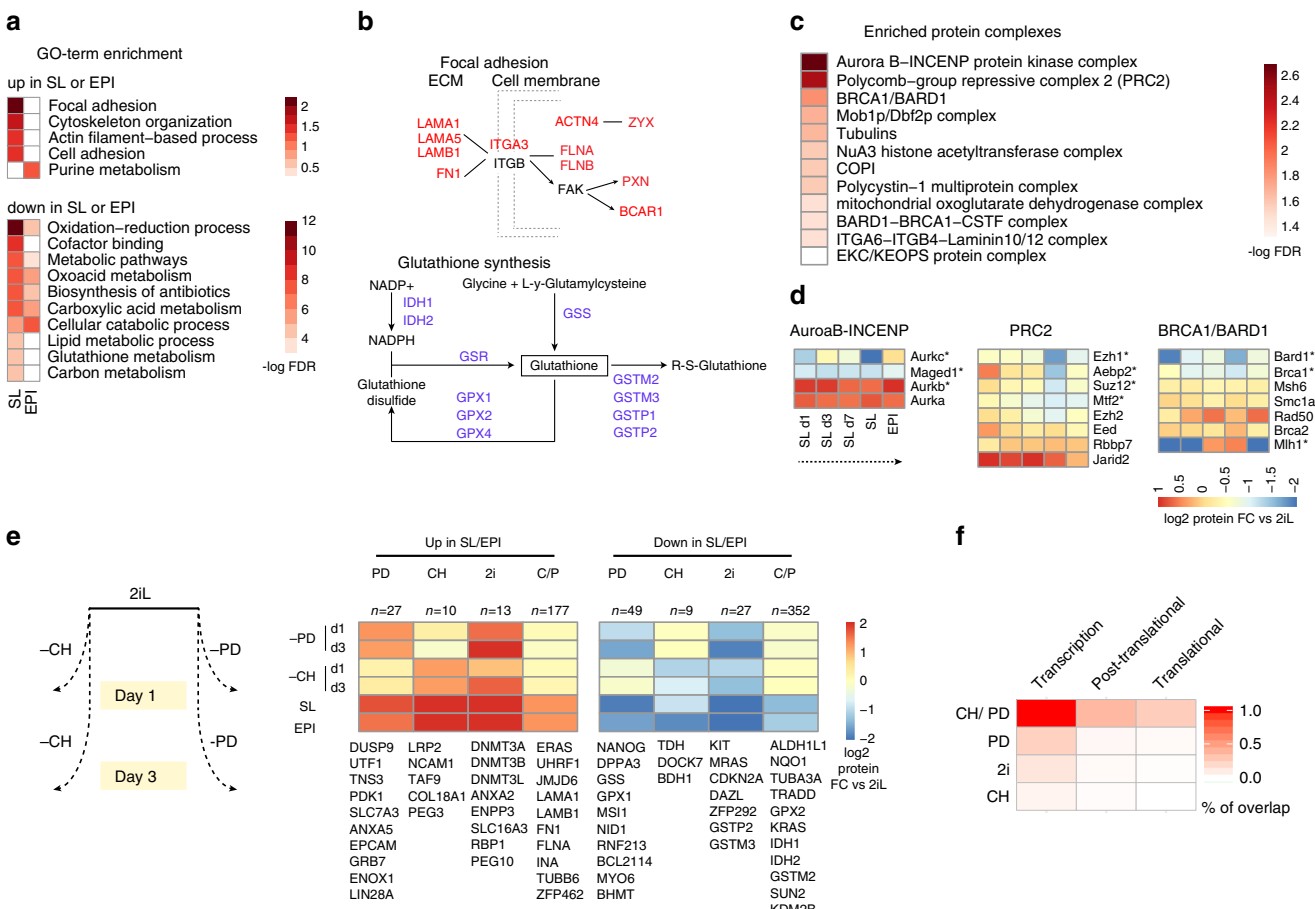

**Fig. 5 Functional annotation of differentially expressed proteins in different states of pluripotency. a** Gene ontology analysis of differentially expressed proteins in SL or EPI when compared to 2iL state. Two-sided P values were adjusted for multiple testing using Benjamin–Hochberg correction. **b** Examples of GO-term categories enriched in SL (focal adhesion) or in 2iL (glutathione metabolism). Differentially expressed proteins belonging to each category are shown. Upregulated proteins in SL or EPI are shown in red and upregulated proteins in 2iL are shown in blue. **c** Enrichment analysis for large protein complexes using differentially expressed genes between 2iL and SL/EPI. Two-sided P values are derived from the hypergeometric test associated with each protein complex and were adjusted for multiple testing using Benjamin–Hochberg correction. **d** Heat-map showing the fold change in protein expression for members of the top three enriched protein complexes from panel **c** and during 2iL to SL and EPI transition. **e** Single inhibitor withdrawal proteomics. 2iL cells were cultured for 1 and 3 days with serum-free medium supplemented with CH/LIF or PD/LIF and were employed in proteome analysis. Heat-map showing the change in protein expression for the 642 differentially expressed proteins between SL/EPI and 2iL. "C/P" refers to the proteins that require either PD or CH to maintain expression similar to 2iL ESCs; only removing both inhibitors from 2iL induces changes resembling the SL/EPI states. "2i" refers to the proteins that require both PD and CH signaling to maintain expression similar to 2iL state; removing either PD or CH from 2iL induces changes similar to SL/EPI state. "P" refers to the proteins that require PD to maintain an expression pattern similar to 2iL. "C" refers to the proteins that require CH to maintain an expression pattern similar to 2iL. n = 642 differentially expressed proteins. **f** The overlap between differentially expressed proteins regulated at the three levels and downstream of PD or CH single inhibitors.

removing either the PD or the CH inhibitors was not sufficient to induce a gene expression pattern that is observed in SL/EPI states (Fig. 5e). This indicates that signaling downstream of either PD or CH is sufficient to maintain a gene expression signature similar to 2iL. On the other hand, 95 proteins significantly changed expression upon PD or CH withdrawal (fold change >2 and FDR < 0.05) and we could accurately assign these proteins to signaling pathways downstream of CH or PD inhibitors. Overall, we observed a greater effect of PD than CH withdrawal on differential protein expression (76 vs. 19 proteins, respectively). Further, for a specific set of proteins (n = 40), we found that the function of both PD and CH is required to induce a gene expression signature similar to 2iL state. These proteins included members of the DNA methyl transferase complex (including DNMT3A, DNMT3B, and DNMT3L) that gained similar expression as SL/EPI when either the PD or the CH was removed from the 2iL culture.

In summary our findings revealed genes that undergo specific transcriptional, translational, or post-translational regulation in ground state ESCs when compared to SL or EPI states (Fig. 5, Supplementary Data 5). We provide a plausible link between the observed differential protein expressions and the individual signaling pathways downstream of PD- or CHIRON-function in ground state pluripotency. We observed that the majority of differentially expressed proteins requires either PD- or CHIRON signaling to maintain their expression similar to 2iL state; for some proteins (e.g. DNMT3B or TET2), however, both inhibitors (2i), are required and removing either PD or CH from the 2iL medium induces changes similar to SL/EPI states. Altogether, by integrating different layers of gene expression and linking these changes to the upstream signaling pathways, we provide a comprehensive and detailed overview of the global changes in gene expression during the naive to primed ESC transition (summarized in Fig. 6, Supplementary Data 5).

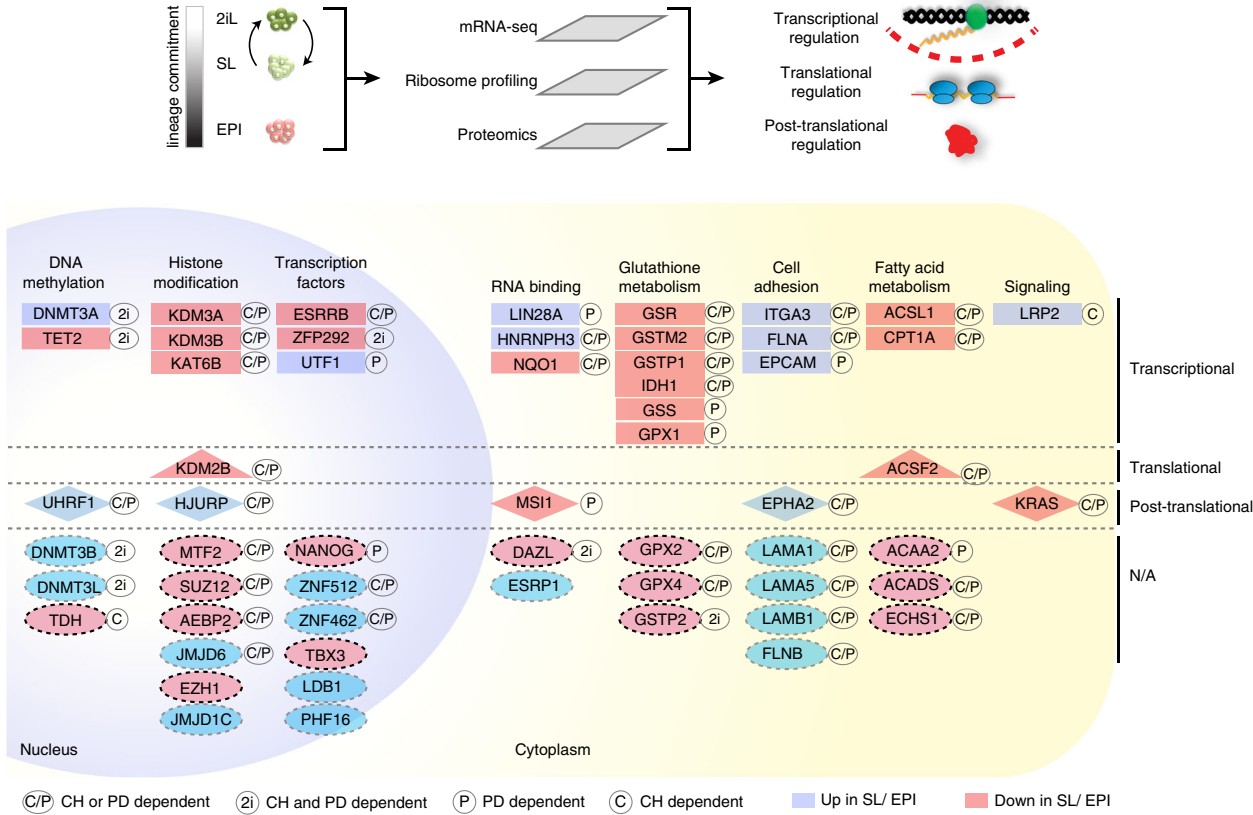

**Fig. 6 Diverse modes of gene regulation in pluripotent cells.** An overview of the multiple layers of gene regulation that have been integrated in this study. Examples of genes that undergo specific regulation at the transcriptional, translational, or post-translational levels are shown. Genes that could not be assigned to any specific mode of gene regulation (e.g. because the RFP, RNA, or protein could not be uniquely integrated in different conditions) are indicated with NA. Genes were also assigned to signaling pathways downstream of CH or PD, as indicated in the figure. "C/P" refers to the proteins that require either PD or CH to maintain expression similar to 2iL ESCs; only removing both inhibitors from 2iL induces changes resembling the SL/EPI states. "2i" refers to the proteins that require both PD and CH signaling to maintain expression similar to 2iL state; removing either PD or CH from 2iL induces changes similar to SL/EPI state. "P" refers to the proteins that require PD to maintain an expression pattern similar to 2iL; removing PD from 2iL medium induces changes similar to SL/EPI state. "C" refers to the proteins that require CH to maintain an expression pattern similar to 2iL; removing CH from 2iL medium induces changes similar to SL/EPI state.

## Discussion

In this study, we provide a comprehensive and detailed picture of the different layers of gene regulation in naive and primed state pluripotency. We started by asking whether and to what extent translational control contributes to differential gene expression and protein levels variation during the naive to primed ESC transition. We found that 2iL state is associated with increased polysome density and TE. This observation cannot be explained by general increase in stem cell differentiation or cell proliferation, both of which are generally associated with augmented mRNA translation and protein synthesis[2,9,36]. In this regard, 2iL ESCs display lower lineage commitment and cell proliferation rate when compared to SL and EPI states[18,35]. Our findings, however, may not contradict the previous observations of a low protein synthesis rate in stem cells as we found a poor correlation between increased ribosome occupancy and enhanced protein expression for the majority of examined RNAs. Thus the translational control contributes minimally to the final protein variation during naive to primed ESC transition.

Conspicuously, the majority of mRNAs with higher TE showed lower mRNA abundance that is compensated by the increased mRNA translation in 2iL ESCs, ultimately leading to similar protein expression levels across different ESC states. This observation highlights the effect of gene expression buffering; a coordinated balancing act between transcriptional and translational machineries that likely provides a mechanism to offset the transcriptional noise and to ensure consistent and precise regulation of protein expression[9]. Cells require robust mechanisms for coping with internal noise (e.g. stochastic initiation of transcription) or fluctuations in external signals to tightly regulate protein expression. In fact, during the cell state transition in mammalian cells, post-transcriptional control accounts for significant delays in induction of changes in the protein levels, despite comparatively fast shifts in the mRNA levels[9,37]. The phenomenon of post-transcriptional buffering at the translational or post-translational levels has been observed in diverse contexts; for instance, studies in tumor cells show that copy number alterations result in less variation in protein levels than the corresponding mRNAs[38,39]. Similarly, genetic variations among individuals (quantitative trait loci, QTLs) have less effect on protein than the corresponding mRNA levels[40], suggesting that the effects of QTLs on the downstream phenotypes are buffered. A similar observation was also made by comparing mRNA and protein levels among primates (human chimpanzee and rhesus macaques), where many genes with significant mRNA differences showed little or no change in protein levels indicating a stronger evolutionary constraint for protein expression[5,41]. Furthermore, specific buffering at the translational level has been observed

between yeast[42] or bacterial species[43], across human individuals[44], in aneuploid tumor cells[45], and in response to growth factor stimulation[46]. Thus, post-transcriptional buffering (both at translational or post-translational levels) can lead to comparable protein abundance and divergence between mRNA and protein expression.

Interestingly, we observed a significant enrichment for RBPs among the translationally regulated mRNAs. RBPs impact various stages of mRNA expression and turnover and play critical roles in stem cell pluripotency and differentiation. Several differentially translated RBPs in our data such as *Mettl14* (ref. [47]), *Pabc1* (ref. [48]), *Pcbp1* (ref. [49]), *Ddx6* (ref. [50]), and *Hnrnpd*[51] have been implicated in the maintenance or differentiation of embryonic and adult stem cells. However, it remains an open question how these RBPs are translationally regulated in different states of ESCs. A potential mechanism affecting some of these mRNAs could be the activity of the ELAVL; ELAVL1/2/3/4 protein family, which are known to specifically bind the AREs. While AREs were demonstrated to both increase or decrease translation and mRNA stability in various contexts[52–55], ELAVL proteins are generally linked to the enhanced stability and translation of their target mRNAs[55–57]. We found a considerable enrichment of AU-rich elements and the consensus-binding motif of the ELAVL proteins in the 3′ UTRs of mRNAs that showed higher TE in 2iL ESCs. Accordingly, we observed a significant downregulation of ELAVL2/4 proteins in both SL and EPI when compared to 2iL state. Thus, downregulation of the ELAVL proteins in SL and EPI cells might be linked to the decreased TE of the ARE-containing mRNAs in these cells. Our data also demonstrate a strong induction of the RBP DAZL in 2iL ESCs. DAZL stimulate mRNA translation in cooperation with the poly(A)-binding protein (PABP)[58]; DAZL was shown to specifically bind to the 3′ UTR of the ELAVL2 mRNA and enhance its stability and TE[59]. The involvement of DAZL and its associate factors in regulation of the expression of ELAVL proteins and their impact on the higher rate of general, as well as transcript-specific, mRNA translation in 2iL cells would be an interesting subject for further investigations.

By comparing the steady-state mRNA level, ribosome occupancy, and protein abundance in the three pluripotent states (i.e. 2iL, SL, and EPI) we showed that changes in mRNA expression can largely explain the changes in protein abundance for ~70% of genes. Conspicuously, we found that translational control accounts for ~10% of protein variations. In a seminal work describing the correlation between mRNA and protein levels, ~40% of differences in protein levels was attributed to variation in mRNA expression suggesting a major impact of translation on controlling the protein levels in mammalian cells[6]. An independent study concluded that only ~9% of variation in protein levels could be attributed to specific translational control and mRNA change explained most of the variation in protein levels[60]. Similarly, in lipopolysaccharide-stimulated dendritic cells, change in mRNA abundance plays a dominant role in determining the changes in protein levels[61]. Accordingly, the inter-species divergence in protein levels was largely explained by changes in RNA levels and the divergence in TE accounted for less than 20% of variation in protein abundance[41]. Thus in specific cellular contexts, change in mRNA abundance can largely explains the variation in protein levels in steady-state measurements[2,9,62,63], while translational and post-translational controls could enable rapid, temporal adaptation during cell state transitions.

Given that we have measured the steady-state mRNA levels, our observed changes in mRNA abundance could be due to either transcriptional regulation or mRNA stability, both of which could lead to changes in the final mRNA abundance. Recently, Fremier et al. investigated the transcriptome-wide changes in mRNA

translation and stability in the context of microRNA-induced gene silencing[64]. In this regard, microRNAs were shown to regulate both the stability and translation of their target mRNAs. Notably, depletion of DDX6, a RNA helicase implicated in microRNA-induced gene silencing, affected mRNA translation without impacting their stability, thus demonstrating a role for translational repression, independent of mRNA destabilization. By measuring nascent mRNA transcription and steady-state mRNA levels, Fremier et al. also employed ESCs maintained in a mix of SL + 2iL culture condition and showed that mRNA transcription rather than change in mRNA stability is the dominant regulator of mRNA abundance during ESCs-to-EPI differentiation. Furthermore, Fremier et al. reported limited differential mRNA translation in the employed ESCs when compared to EPI cells. Our findings corroborate and extend these observations and suggest that transcriptional control represents the main mode of gene regulation among different states of ESCs whereby specific translational and post-translational control play a lesser contribution to regulating the final protein abundance. We also note that for a subset of genes the change in protein abundance cannot be extrapolated by analyzing RNA-seq data alone. We show that this group of genes are regulated mainly at the post-translational level and to a lesser extent at the translational levels.

## Methods

**ESC culture**. For SL culture, E14Tg2a ESCs (purchased from ATCC) were maintained on gelatin-coated dishes without feeder cells and in Dulbecco's modified Eagle's medium (DMEM) supplemented with 10% fetal calf serum, L-glutamine (2 nM), Na-Pyruvate (1 mM), non-essential amino acids (0.1 mM each), penicillin/streptomycin (Gibco), 2-mercaptoethanol (55 μM), and LIF (1000 U/ml, Milipore). For 2iL culture, E14Tg2a ESCs were cultured in NDiff 227 medium (StemCells, Inc.) supplemented with MEK inhibitor PD0325901 (1 μM), GSK3 inhibitor CHIR99021 (3 μM), and LIF (1000 U/ml, Milipore). SL-ESCs were cultured for over 2 months (>24 passages) in 2iL medium to generate the steady-state 2iL ESCs. For EPI differentiation, E14Tg2a ESCs were cultured in NDiff 227 (StemCells, Inc.) supplemented with penicillin/streptomycin, 20 ng/ml Activin-A (R&D Systems), 12 ng/ml bFGF (R&D Systems), and 1% knock-out serum replacement (Gibco) and on Fibronectin (10 μg/ml)-coated dishes at ~15,000 cells per cm² density. For EPI differentiation, 2iL-ESCs were transferred to EPI medium and were maintained for 72 h. For single-inhibitor experiments and to assess the (likely) direct effects of PD and CHIRON signaling, we cultured the steady-state 2iL ESCs for 1 and 3 days in single inhibitor-supplemented medium and collected the cells for further experiments.

**Polysome profiling assay**. Cells were pretreated with 100 μg/ml cycloheximide (Sigma) for 5 min, and lysed in hypotonic buffer (5 mM Tris-HCl (pH 7.5), 2.5 mM MgCl₂, 1.5 mM KCl, 1× protease inhibitor cocktail (EDTA-free; Roche), 100 μg/ml cycloheximide, 2 mM DTT, 200 U/ml RNaseIn, 0.5% (v/v) Triton X-100, and 0.5% (v/w) sodium deoxycholate) to isolate the ribonucleoproteins. Four hundred micrograms of the ribonucleoproteins were separated on a 10–50% sucrose gradient by ultracentrifugation at 36,000 r.p.m. in an SW40 rotor (Beckman Coulter) at 4 °C for 2 h, and fractionated using an ISCO gradient fractionation system and optical density at 254 nm was continuously recorded with a FOXO JR Fractionator (Teledyne ISCO).

**Ribosome profiling assay**. Ribosome profiling was performed as previously described[65], with minor modifications. Briefly, 500 μg of the ribonucleoproteins (two biological replicates per sample) were treated with 1000 U RNase I (Ambion Cat# AM2295) at 4 °C for 50 min with gentle end-over-end rotation followed by incubation with SuperaseIn (Ambion, Cat# AM2696). Monosomes were pelleted by ultracentrifugation in a 34% sucrose cushion at 70,000 r.p.m. in a TLA-120.2 rotor (Beckman Coulter) at 4 °C for 3 h. RNA fragments were extracted with acid phenol (2×), once with chloroform, and precipitated with isopropanol at −20 °C in the presence of NaOAc and GlycoBlue (Invitrogen). Purified RNA samples were resolved on a denaturing 15% polyacrylamide-urea gel and the sections corresponding to 28–32 nucleotides containing the RFPs were excised, eluted, and precipitated by isopropanol.

Purified RFPs were dephosphorylated using T4 polynucleotide kinase (New England Biolabs) for 1 h at 37 °C. Denatured fragments were re-suspended in 10 mM Tris (pH 7) and quantified using the Bio-Analyzer Small RNA assay (Agilent). A sample of 10 pmol of RNA was ligated to the 3′-adaptor with T4 RNA ligase 1 (New England Biolabs) for 2 h at 37 °C. Reverse transcription was carried out using oNTI223 adapter (Illumina) and SuperScript III reverse transcriptase

(Invitrogen) according to the manufacturer's instructions. Products were separated from the empty adaptor on a 10% polyacrylamide Tris/Borate/EDTA (TBE-urea) gel and circularized by CircLigase (Epicentre). Ribosomal RNA amounts were reduced by subtractive hybridization using biotinylated rDNA complementary oligos[65]. The RFP libraries were amplified by PCR (11 cycles) using indexed primers and quantified using the Agilent BioAnalyzer High-Sensitivity assay. DNA was then sequenced on the HiSeq-2000 platform with read length of 50 nucleotides (SR50) according to the manufacturer's instructions, with sequencing primer oNTI202 (5CGACAGGTTCAGAGTTCTACAGTCCGACGATC).

**RNA-seq assay.** RNA isolation was performed using two biological samples per time point and the Qiagen RNeasy protocol. Four micrograms of RNA was employed to generate the RNA-seq libraries. In short, total RNA was subjected to rRNA depletion using the Ribo-ZeroTM Gold kit (Epicentre) according to the manufacturer's instructions. rRNA-depleted samples were employed for first- and second-strand cDNA synthesis and using dUTP for strand-specificity. RNA-seq libraries were prepared using KAPA Hyperprep kit (Roche 07962363001) according to the manufacturer's instructions and the final libraries were sequenced on a HiSeq-2000 Illumina sequencer.

**Analysis of ribosome profiling and RNA-seq data.** To analyze the ribosome profiling data, FASTQ reads were processed as previously described[65]. Briefly, adaptor sequences were removed using fastx_clipper (fastx_toolkit-0.0.14) (http://hannonlab.cshl.edu/fastx_toolkit/index.html) and by employing the following parameters clipper -Q33 -a CTGTAGGCACCATCAAT -l 25 -c -n -v. Reads were then trimmed using fastx_trimmer (fastx_toolkit-0.0.14) and -Q33 -f 2 parameters. Both RNA-seq reads and the trimmed RFP reads were then aligned against mouse rRNA, tRNA, snRNA, snoRNA, mtRNA sequences using Bowtie[66] and -seedlen=23 to deplete these small RNA contamination. Unmapped reads (cleaned) were then mapped against mouse genome (mm9) using GSNAP[67] and by employing the following parameters: -B 5 -t 15 -N 1 -E 100 -w 100000 -n 10 -s mm9refGene_splice. Mapped reads were counted in the coding sequence using HTseq-count[68] with the settings: -m union -s reverse -t CDS.

RFP and RNA-Seq read counts were normalized for quantification of the TE quantification and differentially translation analysis using Xtail algorithm v.1.1.5 (ref. [69]) with minMeanCount=50. Genes were considered differentially translated if they had at least 50 normalized reads in RNA, 25 normalized reads in RFP, displayed absolute fold change >1.5 and FDR ≤ 0.1. The DESeq package[70] was also used to normalize the RFP and RNA-Seq read counts (with similar results to Xtail normalization) and the results were employed to correlate the variation in RNA, RFP, and protein levels. Graphs were generated using data.table and ggplot2 R packages in R version 3.5.1 on Ubuntu 16.04.5 LTS.

**Quality control of Ribo-Seq data.** To analyze the position of initiating ribosome in relation to the annotated translation initiation sites (TISs), we employed a custom pipeline as reported previously[52]. Briefly, BAM files were converted to strand specific 5′ end wiggle files using a custom Pyton script. Wiggle files were then converted to a format suitable for the Batch PositionConverter Interface in Mutalyzer 2.0.beta-32 (https://mutalyzer.nl/batchPositionConverter). These converted files were then manually loaded into Mutalyzer to retrieve positions relative to the annotated TIS. We analyzed the first position of the aligned reads to transcript coordinates and relating those coordinates to annotated TISs positions located up to −15 nt surrounding the TIS were counted as positions in coding regions For all samples, a major peak was observed at −12 nt from the annotated TIS.

To calculate the read distribution and coverage at 5′ UTR, CDS, and 3′ UTR regions, we first downloaded all the 5′ UTR, CDS, and 3′ UTR sequences from Ensemble BIOMART mm9 database. CDS sequences were extended by 50 nt upstream for 5′ UTR analysis and by 200 nt downstream for 3′ UTR analysis. We used CDS sequences with more than 1000 nt for this analysis. The generated BED file for 5′ UTR-CDS or 3′ UTR-CDS were binned into 10 nt bins and the number of reads from RNA-seq and RFP-seq libraries were counted using BEDtools, Samtools and in house script (peakstats.py).

**Proteome profiling with LC-MS/MS.** To generate whole-cell proteomics profiles, cells were lyzed in 4% SDS, 0.1 M HEPES pH 7.6, 0.1 M DTT. Protein lysates were then digested with trypsin/LysC combination using Filter Aided Sample-Preparation (FASP) as described previously[71]. Peptides were desalted and purified on StageTips[72] and were loaded on Orbitrap LC-MS (Thermo). Two set of proteomes were generated: the forward proteome (included 2iL, SLd1, SLd3, SLd7, SL, and EPI samples) and the reverse transition (that included SL, 2iLd1, 2iLd3, 2iLd7, and 2iL samples). MaxQuant 1.5.1.0 (ref. [31]) was used to analyze the mass spectra. Briefly Thermo Raw MS files were used to search against the curated mouse RefSeq protein sequence database using default MaxQuant settings with match-between-runs, and label-free quantification (LFQ) and iBAQ quantification of proteins enabled[31,73]. One percent false-discovery rate was applied to the match of propensity-score matching and assembly of proteins. Two missed cleavages were allowed for trypsin enzyme cuts and peptides length was set between 1 and 7 amino acids. Perseus software (version 1.5.5.3) was used to perform filtering, imputation of missing values, and permutation-based t-test. In brief, the generated LFQ and IBAQ values from MaxQuant were used

and the identified proteins were searched against a decoy database from MaxQuant. We filtered out proteins that flagged as "reverse" or "contaminant" from the final list. To calculate differential protein expression, biological triplicates were grouped and the protein list was filtered for proteins that were not reproducibly quantified in three replicates in at least one conditions of the forward or reverse proteome samples. Next, missing values were imputed from a normal distribution using the default settings (width = 0.3, down shift = 1.8). Lastly, differential proteins between triplicates were calculated using a Student's t-test (FDR < 0.05) and a fold-change of >3-fold, following previous recommendations[74].

**qRT-PCR.** To validate the candidate genes expression in polysome profiling assay, sucrose gradient fractions were used for RNA extraction by using TRIzol (ThermoFisher). RNA samples were employed in cDNA synthesis using 1 μg of RNA according to the manufacturer's instructions. Fractions collected from two independent experiments were used. Quantitative PCR was performed using iQ SYBR green supermix (Bio-Rad) and primers in Supplementary Data 6. Ct values were normalized to input samples from un-fractionated (total) lysates.

**Western blot analysis.** ES cells were lysed in RIPA buffer supplemented with protease inhibitors (11836170001; Roche). Twelve percent SDS-PAGE gels were used for protein separation followed by transfer onto PVDF membrane. Blots were blocked with 5% non-fat milk at room temperature for 1 h and incubated with the following primary antibodies overnight at 4 °C: RNF126 (Abcam, ab234812, 1:500), BHMT (Abcam, ab96415, 1:500), NQO1 (Abcam, ab28947, 1:500), KRAS (Abcam, Ab180772, 1:500), S100A6 (Abcam, ab134149, 1:1000), OCT4 (Millipore, MABD76, 1:1000), and ESRRB (Perseus proteomics, PP-H6705-00, 1:1000). Primary antibodies for GAPDH (Abcam, Ab8245, 1:1000) and ACTB (Sigma, A1978, 1:1000) were employed as internal controls and for 1 h at room temperature. HRP-swine-anti-rabbit (Dako, P0217) and HRP-rabbit anti mouse (Dako, P0161) were used as secondary antibodies and subsequently signal was detected using ECL kit (Pierce, 32106) and the ImageQuant LAS 4000 system (GE Healthcare Life Sciences). Western blots were repeated for two times.

**Comparison with embryonic profiles.** We first compared the profiles of 2iL, SL, and EPI ESCs and selected genes that are specific for each state or shared between SL and EPI when compared to 2iL state (FC ≥ 3 and DEseq, adjusted P value <0.05). We then performed hypergeometric enrichment analysis and computed the enrichment of ESCs-specific genes within stage-specific gene sets previously reported for different stages of the developing mouse embryo[30]. All P values were corrected using Benjamini–Hochberg test.

**UTR analysis.** We retrieved the 5′-UTR, CDS and 3′-UTR sequences from Ensemble BIOMART and used the full-length sequences for structural features analysis (including nucleotide composition, sequence enrichment, and motif enrichment analysis). To analyze the sequence around the start and stop codons, we selected the 50 nt flanking the start site and the 200 nt flanking the stop codon. To compute the length and the %GC of different sequences the bedtools nuc from BEDTools suite v2.20.1 was used. To compute the AU-rich elements (ARE) enrichment, we scanned for the UAUUUAU elements in the 3′ UTR regions of differentially translated mRNAs (n = 108), all mRNAs that were included in TE-analysis (n = ~6300) and randomly selected 3′ UTR regions (n = 108) using Gimme scan and the following parameters: -c 0.9 –r. AREs were scanned in the positive strand of 3′ UTRs and mRNAs with at least one ARE in their 3′ UTR region were counted. Hypergeometric test was performed to compute the enrichment of AREs in differentially translated genes and random set of genes.

**RBP motif analysis.** We used the CISBP-RNA database[75] that includes RNA motifs for 228 RNA-binding proteins. Gimme Motifs package scan[76] was then used with –c 0.9 –r to scan all motifs for the known RNA-binding proteins in the 3′ UTR of differentially translated genes, total genes (6K), and 108 randomly selected genes. We captured ~186 unique RBP motifs in DE genes, all genes, and in random list. Hypergeometric test was then performed to calculate the enrichment for different RBP-motifs for the differentially translated mRNAs and the random set of genes.

**Protein complex enrichment analysis GO-term analysis.** To compute the enrichment of different protein complexes in the list of differentially expressed proteins, we used the recently annotated compendium of 275 large protein complexes (>5 members)[33]. Hypergeometric test was used to calculate the enrichment and P values were adjusted for multiple testing using Benjamini–Hochberg correction and protein complexes with FDR ≤ 0.1 were selected.

**Statistics and reproducibility.** R version 3.5.1 on Ubuntu 16.04.5 LTS was used for statistical analyses. Error bars, P values, and statistical tests are reported in the figure legends. Biovenn was used to generate the venn diagram for overlap between RNA, RFP, and proteins. http://www.biovenn.nl/. Statistical tests include paired or unpaired two-tailed Student's t-test, Fisher's exact test, Wilcoxon rank-sum test, "N−1" Chi-squared test, Pearson correlation, and Wald test. All experiments were performed independently at least two times unless otherwise indicated.

**Reporting summary**. Further information on research design is available in the Nature Research Reporting Summary linked to this article.

## Data availability

All high throughput datasets have been deposited at Gene Expression Omnibus (GEO) with accession code GSE133794. The mass spectrometry data have been deposited at the ProteomeXchange Consortium via the PRIDE partner repository with the dataset identifier PXD014528. The source data underlying Figs 1–6 and Supplementary Figs 1–4 are are provided as a Source Data file.

## Code availability

Codes are available as part of the source data file

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

## Acknowledgements

We thank Nina C. Hubner, Pascal W.T.C. Jansen, and Luan N. Nguyen for assistance with proteomic analysis; Ehsan Habibi for assistance with bioinformatics analysis; Eva Janssen-Megens, Kim Berentsen, and Nilofar Sharifi for sequencing; Colin Logie and Hendrik Marks for informative discussions; and Angela Hackett for technical assistance. The computational work was carried out on the Dutch national e-infrastructure with the support of SURF Cooperative. SMJ is supported by the Patrick Johnston research fellowship from Queen's University Belfast. MV is supported by the Oncode Institute, which is partly funded by the Dutch Cancer Society (KWF). This research was supported and funded by the European Union grant ERC-2013-ADG-339431 "SysStemCell".

## Author contributions

Y.A., S.M.J., N.S., and H.G.S. conceived the study and wrote the paper which was reviewed by all authors. Y.A. and S.M.J. performed the experiments. Y.A. analyzed the data. C.G.G. assisted with the ribosome profiling and M.V. assisted with proteomic and mass spectrometry experiments.

## Competing interests

The authors declare no competing interests.
