## [Peer Review File · Nature Communications]

Reviewers' comments:

Reviewer #1 (Remarks to the Author):

Translational regulation of gene expression plays critical roles during early embryonic development, although the translational landscape of early mammalian development remains to be under-studied. Atlasi et al. addressed this important question by exploring the dynamics of mRNA, translation and protein abundance during the interconversion between naïve and primed pluripotent states. They employed genome-wide approaches, including RNA-seq, ribosome profiling, polysome profiling, and mass spectrometry, combined with time-course analysis of their dynamics. They showed that a strong translational buffering of a specific set of mRNAs in 2iL ESCs leads to stable protein levels and that the global change of proteome is largely accompanied by transcriptional rewiring. They also pointed out that the post-translational regulation is important for specific genes during the transition of naïve into primed pluripotency.

This manuscript provides a comprehensive and detailed genome-wide landscape of transcription, translation and protein abundance during the transition of naïve to primed pluripotency, which is insightful and resourceful to better our understanding early development and stem cell pluripotency. However, major and minor concerns exist related to the data quality and analysis, described in detail below, and should be addressed to further improve the manuscript.

Major points:

- 1, Fig 1b: The polysome profiling of SL looks abnormal: the highest peak is at 60S instead of 80S and polysome part is decreased too much-it looks like the translation machinery is cracked in these cells. This is an important result in this paper that forms the foundation of the rest of results / analyses/ conclusions in the paper.
- 2, One important information is missing for the data in Fig 1a to 1f: Are these cells at their steady states? Or how long the cells were cultured in SL/2iL/EPI after the transfer from the original condition? From Fig 1g, the authors know the importance of the transition timeline and make sure whether the cells are at steady state. But in "Embryonic stem cell culture" part of "Methods", it is said "Samples were collected after 1 or 3 days of culture". What does this mean?
- 3, The label for EPI cell state should be consistent, for example: in introduction, it is "EpiSCs" or "EpiLSCs"; in Fig 1, it is "EPI"; in Fig 2, it is "EpiLSCs"; in Fig 2 legend, it is "EPI".
- 4, How did the authors compare the protein abundance among different samples from their mass spectrometry assays? The authors should give the analysis details and provide minimal validation results to show the comparison of protein abundance among different samples, like WB results of a few candidates in both 2iL/SL/EPI and SL-2iL transition.
- 5, Fig 2e: The FC of Rnf126 protein levels are all more than 1.5 in serD1-7 and EpiLSCs, but FC in serum is close to 1. The author should give the timeline of transition to show how long the cells need to get the steady state from 2iL to SL. From Fig 2e, it looks like at D7, the cells were still undergoing active regulatory adjustment. But from Fig 1g, the most dramatic change happens at the early transition time D1, while the condition at D7 is similar to that in SL.
6. Fig 5 model should be explained and described in more detail rather than just being mentioned in the last sentence of the Results part.

Minor points:

- 1, Fig 1b: The top: the sizes of the free subunits (40S and 60S) should be the same as the ones in the following monosome / polysomes.
- 2, Fig 1b, Figure legend: polysome instead of "Ploysome".
- 3, Fig 1b: For the density of polysome profiling, it is usually categorized into 40S, 60S, monosome (or free subunits and monosome, or prepolyosome), light polysomes, heavy polysomes. It would be better to follow these categories.
- 4, Page 5 in the middle- A typo: "In contrast to SL culture, both 2iL and EPI cultures are based on serum-free medium supplemented with small molecule inhibitors". It should be "molecule".
- 5, Fig 1f: in Fig 1d, common vs 2iL shows that there are 28 genes having higher TE in SL/EPI. But in Fig 1f and Fig S3c, there are 30 genes. The author should give some explanations for the discrepancy.
- 6, Page 6: subtitle: "Integrative analysis of ribosome occupancy, RNA expression, and protein expression suggest pervasive translational buffering in ground state pluripotency". It should be "suggests".
- 7, Fig 3d legend: "Transcriptional: protein change >3 fold and RNA change >2 fold". It should be "RFP change > 2 fold" based on the text on page 8.
8. Page 2: (DNA>RNA>protein) is better presented as (DNA↔RNA↔protein)
9. I may have overlooked, but it seems that the data for Fig 3e and 3f were not mentioned/referred to in Results?
10. Fig 4e: I found it difficult to understand the right part of the illustration with PD, CH, 2i, CH/PD symbols pointing with --| or black ◊. May need add some explanations in Fig legends. Also, aren't 2i the same as CH/PD?

Reviewer #2 (Remarks to the Author):

Atlasi and Jafarnejad et al characterized how gene expression changes at different levels during early mouse embryonic stem cell differentiation. They employ a suite of high-throughput approaches including ribosome profiling, proteomics, and RNA sequencing to measure gene expression at the transcriptional, translational, and protein levels. They find that most changes in ultimate gene levels are driven by transcription, and find a buffering role for translational control to maintain protein levels between cell states. It is not clear how cells could achieve buffering at the translational level for an arbitrary gene set that changes in transcription but not protein level, and it makes me wish for attempts to answer this. Is the gene set enriched in particular regulatory elements? How might cells accomplish this? Without this the work ends up being rather descriptive and leaves next steps unclear. The work would be strengthened considerably by testing the hypothesis that translation is buffering transcriptional changes in this system.

Major comments:

Throughout the paper the authors make general claims about the behavior of cells as they

differentiate. It is unlikely that differentiation between two arbitrary cell types is regulated identically. As such, it is critical that the authors more precisely qualify their claims about differentiation to reflect the differentiation they are studying: ground-to-primed embryonic stem cells.

Translation efficiency (TE) is a noisy metric, as any noise from RNAseq in the denominator is amplified when taking the ratio. The authors also identify a set of genes on page 8 which have a change in RFP of >2 fold but no change in RNA (< 2 fold). Is this set of genes that uniquely change in translation level a better predictor of protein levels than TE?

Please clarify in the methods what the origin of 5' and 3' UTR sequences is: in one section it is stated that they are defined as 50 nt upstream and 200 nt downstream of the CDS, while in another section it says they were obtained from UCSC BIOMART (which itself is confusing since I normally associate BioMart with Ensembl).

Minor points:

The authors should indicate the organism they have performed the experiments in in the abstract.

I am not sure that "the translational landscape of early mammalian development ... remains largely unexplored". The authors may wish to consider replacing this statement with a more precise statement of the gap they seek to fill, as there are hundreds (or maybe thousands) of studies on translational control in early mammalian development. A similar sentence exists on the bottom of page 3. Specifically, the authors should present their results in light of Freimer et al 2018, who studied how translation and mRNA half-lives change as mouse ESCs differentiate.

The authors may wish to clarify what they are referring to as "82% and 81%" etc on page 6, as from the text it seems they are discussing correlation coefficients and not percents.

AU-rich elements can both increase or decrease translation and mRNA stability (which are also linked). The authors assert on page 11 that ARE's are "generally linked to enhanced stability and translation" and it would be nice if this statement was supported by a reference.

Reviewers' comments:

Reviewer #1 (Remarks to the Author):

Translational regulation of gene expression plays critical roles during early embryonic development, although the translational landscape of early mammalian development remains to be under-studied. Atlasi et al. addressed this important question by exploring the dynamics of mRNA, translation and protein abundance during the interconversion between naïve and primed pluripotent states. They employed genome-wide approaches, including RNA-seq, ribosome profiling, polysome profiling, and mass spectrometry, combined with time-course analysis of their dynamics. They showed that a strong translational buffering of a specific set of mRNAs in 2iL ESCs leads to stable protein levels and that the global change of proteome is largely accompanied by transcriptional rewiring. They also pointed out that the post-translational regulation is important for specific genes during the transition of naïve into primed pluripotency.

This manuscript provides a comprehensive and detailed genome-wide landscape of transcription, translation and protein abundance during the transition of naïve to primed pluripotency, which is insightful and resourceful to better our understanding early development and stem cell pluripotency. However, major and minor concerns exist related to the data quality and analysis, described in detail below, and should be addressed to further improve the manuscript.

We thank the reviewer for pointing out that our study provides “...a comprehensive and detailed genome-wide landscape of transcription, translation and protein abundance” and that the work is “...insightful and resourceful to better our understanding early development and stem cell pluripotency”. We also appreciate the reviewer’s suggestions, which has improved the revised version of this manuscript.

Major points:

1, Fig 1b: The polysome profiling of SL looks abnormal: the highest peak is at 60S instead of 80S and polysome part is decreased too much-it looks like the translation machinery is cracked in these cells. This is an important result in this paper that forms the foundation of the rest of results / analyses/ conclusions in the paper.

We thank the reviewer for highlighting this point. To address this concern and also exclude the potential technical artifacts in our polysome profiling assay, we repeated the experiment with 2 additional replicates using independent batches of cells and experimental setting (as one of the authors relocated to a different lab during the paper revision). We obtained virtually identical results; in all three independent experiments (and in both biological duplicates per experiment, **following figure a**); we observed:

- a) Higher polysome/monosome (heavy fractions/light fractions) ratio in 2i compared with SL.
- b) Serum-ESCs show less 80s and more 40S & 60S peaks when compared to 2i-ESCs.

a) **Low polysome fractions in SL:** we wish to note that a low ratio of heavy polysomes/monosome in ESCs grown in the serum has repeatedly been observed by other groups in the past. This is in fact a known feature of ESCs and other type of stem cells and at least partially is explained by the lower levels of mTOR activity in ESCs (Sampath et al. Cell Stem Cell 2008, doi:10.1016/j.stem.2008.03.013; reviewed in Tahmasebi et al Front Genet 2018, doi:10.3389/fgene.2018.00709).

[Redacted]

To further confirm this point in our experimental setting, we performed new polysome profiling in MEFs and SL and compared the patterns. As expected, we found that serum-ESCs consistently show a lower polysome fraction, even when compared to MEFs (**following figure b**), indicating the low level of mRNA translation in SL compared with differentiated cells.

b) The highest peak is at 60S instead of 80S: We wish to note that we have employed several cell types in our polysome profiling experiments (2iL-ESCs, EpiLSCs and MEFs) and we find a consistent and sharp 80S peak corresponding to vacant ribosomes in all these cells, supporting the reproducibility and technical robustness of the assay. However, it is only in the SL ESC that we observe that, while all peaks are well-defined, the size of 80S is smaller compared with 60S. We believe this phenomenon is likely due to experimental condition that is the standard operation procedure for polysome profiling. The procedure results in a dissociation of the vacant 80S ribosomes into their 40S and 60S subunits without significant impact on the heavy polysomes fraction. Importantly, in all the performed experiments, the heavy fraction/light fractions ratio remains consistently higher in 2iL when compared to SL. To further demonstrate this point, we performed additional assays with different experimental conditions as follows:

- We find that even when a lower amount (150 ug) of the same SL-ESCs lysate is used, there is increase in the vacant 80S peak compared to the higher amount of SL-ESCs (300 ug), without a significant impact on the heavy polysome/monosome ratio. This indicates that the dissociation of the vacant 80S ribosome into 40S and 60S in SL ESCs could depend on the amount of starting material within the same experimental set up (**following figure c**)
- Increasing the salt concentration (KCl) in the sucrose gradient by 5-fold to 500 mM leads to strong dissociation of the vacant 80S ribosome into 40S and 60S subunits in 2iL state, as was reported before in other cell types (*Yoshikawa et al, eLIFE 2018, doi: 10.7554/eLife.36530; Martin and Hartwell, J Biol Chem 1970; van den Elzen et al, EMBO J. 2014, doi: 10.1002/embj.201386123*). Nevertheless, the heavy polysomes fraction still remains higher when compared to regular salt SL ESCs (**following figure d**).
- Conversely, decreasing the salt concentration (20 mM KCl) in the sucrose gradient (5x less), leads to stronger association of the free 40S and 60S subunits and increase in the 80S peak. In these low salt gradients, we observed that the SL ESC still had a lower polysome/monosome ratio compared with 2iL ESCs (**following figure e**), further confirming that the formation of the vacant 80S ribosome is an inherent variation of the experimental set up without affecting the important polysome/monosome ratio.
- A relatively lower 80S peak and fluctuation in its ratio compared to free 60S and 40S subunits have also been observed in previous reports of SL ESCs (**following figure litterateur**). Notably, this phenomenon is not restricted to SL-ESC. For instance, polysome profiling with MCF10a and MCF7 breast cancer cells in multiples studies showed a higher 40S and 60S peaks than 80S (*Dieudonné et al BMC Genomics 2015, doi: 10.1186/s12864-015-*

2179-8; Chaudhury et al, *Nat Commun.* 2016 doi: 10.1038/ncomms13362). Similarly, higher 40S and 60S peak compared with 80S peak was observed in HeLa cells, which have a very active global translation rate as assessed by heavy sedimenting polysome fractions (Yoshikawa et al, *eLIFE* 2018, doi: 10.7554/eLife.36530). Therefore, the presence of 40S and 60S peaks being larger than the 80S peak does not represent a broken translation system and is likely due to technical variations in polysome profiling assays.

Altogether, it appears that the dissociation of vacant 80S ribosomes into 60S and 40S subunits can fluctuate dependent on the experimental setup, without a significant bearing on the heavy polysomes/monosomes ratio, which is the main indicator of mRNA translation efficiency. Importantly, the heavy polysomes ratio, that is the focus of our study, remains consistently higher in 2iL when compared to SL in all the examined conditions, further validating our conclusion for the 2iL and SL comparison.

literature

E14 mESCs grown in 15% SL
Singh S et al PLOS one 2014
doi.org/10.1371/journal.pone.0089098

E14 ESCs grown in 15% SL
Zhang D et al Genes & Dev 2012
(doi/10.1101/gad.182998.111)

iPSCs grown in 15% SL
Tahmasebi S et al Cell Stem Cell 2014
doi.org/10.1016/j.stem.2014.02.005

[Redacted]

2, One important information is missing for the data in Fig 1a to 1f: Are these cells at their steady states? Or how long the cells were cultured in SL/2iL/EPI after the transfer from the original condition? From Fig 1g, the authors know the importance of the transition timeline and make sure whether the cells are at steady state. But in “Embryonic stem cell culture” part of “Methods”, it is said “Samples were collected after 1 or 3 days of culture”. What does this mean?

The E14 ESCs were originally derived and maintained in serum-supplemented medium (SL-ESCs). We cultured these ESCs for over 2 months (>24 passages) in 2iL-medium and generated the steady state 2iL-ESCs. Using LC-MS and bisulfite sequencing we previously measured the global DNA methylation, that is a sensitive marker for ES-state transition, during SL-to-2iL transition and we showed that the global DNA methylation level reaches its minimal and steady state level after ~10 days of SL-to-2iL transition. This steady state level resembles the DNA methylation level observed in the early epiblast of developing mouse embryo (*Habibi et al Cell Stem Cells 2013, DOI: 10.1016/j.stem.2013.06.002, Von Meyenn et al Mol Cell 2016, DOI: 10.1016/j.molcel.2016.04.025*). For EpiLSCs differentiation, we cultured the steady state 2iL-ESCs for 72h in FGF2/ACTIVIN-supplemented medium as previously described. This differentiation protocol is a widely employed approach for EpiLSC differentiation and has been described in detail in several studies (*reviewed in Atlasi and Stunnenberg Nat Rev Genet 2017, doi:10.1038/nrg.2017.57 and Weinberger et al, Nat Rev Mol Cell Biol 2016, doi:10.1038/nrm.2015.28*).

The “*Samples were collected after 1 and 3 days of culture*” refers to the single-inhibitor experiment and not the steady state ESCs. In this experiment, we aimed at studying the individual and (likely) direct effects downstream of PD or CHIRON signaling. We therefore collected cells that are treated for 1 or 3 days in single inhibitor-supplemented medium (starting from 2iL ESCs). As the single inhibitor supplementation is not sufficient to sustain long-term ES-self-renewal we did not collect cells at longer time points as this would have introduced differentiation and secondary effects.

To make this point clearer, we have now added the following text in the methods section:

“SL-ESCs were cultured for over 2 months (>24 passages) in 2iL medium to generate steady state 2iL ESCs. For EpiLSC-differentiation, 2iL-ESCs were transferred to FGF2/ACTIVIN-supplemented medium and were maintained for 72h.” and “For single-inhibitor experiments and to assess the (likely) direct effects of PD and CHIRON signaling, we cultured the steady-state 2iL ESCs for 1 and 3 days in single inhibitor-supplemented medium and collected the cells for further experiments.”

3, The label for EPI cell state should be consistent, for example: in introduction, it is “EpiSCs” or “EpiLSCs”; in Fig 1, it is “EPI”; in Fig 2, it is “EpiLSCs”; in Fig 2 legend, it is “EPI”.

We apologize for the discrepancy in nomenclature; we have now corrected all abbreviations consistently in the text and corresponding figures.

4, How did the authors compare the protein abundance among different samples from their mass spectrometry assays? The authors should give the analysis details and provide minimal validation results to show the comparison of protein abundance among different samples, like WB results of a few candidates in both 2iL/SL/EPI and SL-2iL transition.

We have now extended our description of the MS-analysis in the methods section as follows:

“Two set of proteomes were generated: the forward proteome (included 2iL, SLd1, SLd3, SLd7, SL and EPI samples) and the reverse transition (that included SL, 2iLd1, 2iLd3, 2iLd7 and 2iL samples). MaxQuant 1.5.1.0 [7] was used to analyze the mass spectra. Briefly Thermo Raw MS files were used to search against the curated mouse RefSeq protein sequence database using default MaxQuant settings with match-between-runs, and label-free quantification (LFQ) and iBAQ quantification of proteins enabled. (Cox and Mann, 2008; Cox et al., 2014). 1% false-discovery rate was applied to the match of propensity-score matching and assembly of proteins. Two missed cleavages were allowed

for trypsin enzyme cuts and peptides length was set between 1 and 7 amino acids. Perseus software (version 1.5.5.3) was used to perform filtering, imputation of missing values, and permutation-based t test. In brief, the generated LFQ and IBAQ values from MaxQuant were used and the Identified proteins were searched against a decoy database from MaxQuant. We filtered out proteins that flagged as 'reverse' or 'contaminant' from the final list. To calculate differential protein expression, biological triplicates were grouped and the protein list was filtered for proteins that were not reproducibly quantified in three replicates in at least one the conditions of forward or reverse proteome samples. Next, missing values were imputed from a normal distribution using the default settings (Width = 0.3, Down shift = 1.8). Lastly, differential proteins between triplicates were calculated using a Student's t test (FDR < 0.05) and a fold-change of > 3 fold, following previous recommendations (Pascovici et al., 2016)."

To validate the proteomic data, we have now performed western blot experiments using independent culture of ESCs in SL, 2iL and Epi and during 2iL to SL transition (day1, 3, 7) and SL to 2iL transition (Day1, 3, 7). We examined candidate proteins with available antibodies that included: three 2iL-specific proteins, BHMT, NQO1 and KRAS; a SL specific protein: S100A6; SL and 2iL-ESC-specific protein: ESRRB and lastly a general pluripotency (SL, 2iL and EPI) specific protein OCT4. The western blot data largely confirmed the proteomic data for all the examined proteins, confirming the proteome analysis and quantification. This new data is now added as **Fig 4** in the manuscript.

5, Fig 2e: The FC of Rnf126 protein levels are all more than 1.5 in serD1-7 and EpiLSCs, but FC in serum is close to 1. The author should give the timeline of transition to show how long the cells need to get the steady state from 2iL to SL. From Fig 2e, it looks like at D7, the cells were still undergoing active regulatory adjustment. But from Fig 1g, the most dramatic change happens at the early transition time D1, while the condition at D7 is similar to that in SL.

As the reviewer correctly pointed out, in Fig 1g we show that the major change in TE takes place already within one day of SL to 2iL transition. In this figure, we show the aggregated pattern for ~138 genes during 2iL-to-SL transition. In contrast, in Fig 2e, only one of these genes (i.e. *Rnf126*) is shown. For this particular gene, we also observed a strong increase in RFP level within the 1st day of 2iL to SL transition which results in significant increase in protein level; the upregulation of RFP and protein levels remains evident in all the subsequent time points of 2iL to SL transition. It appears that there are some changes in the absolute levels of mRNA, RFP and protein expression between SL d7 and steady state SL cells. However, we wish to note that the scale in y-axis is LOG2 fold change and therefore the protein level for RNF126 in SL state is still increased by ~2-fold when compared to 2iL ESCs.

6. Fig 5 model should be explained and described in more detail rather than just being mentioned in the last sentence of the Results part.

As suggested, we have further explained Fig 5 and added the following text:

"In summary our findings revealed genes that undergo specific transcriptional, translational or post-translational regulation in ground state ESCs when compared to SL or EPI states (Fig. 5 and Table S5). We provide a plausible link between the observed differential protein expressions and the individual signaling pathways downstream of PD- or CHIRON-function in ground state pluripotency. We observed that the majority of differentially expressed proteins requires either PD- or CHIRON-signaling to maintain their expression similar to 2iL state; for some proteins (e.g. DNMT3B or TET2), however, both inhibitors (2i), are required and removing either PD or CH from the 2iL medium induces changes similar to SL/EPI states. Altogether, by integrating different layers of gene expression and linking these changes to the upstream signaling pathways, we provide a

comprehensive and detailed overview of the global changes in gene expression during the naïve to primed ESC transition”

Minor points:

1, Fig 1b: The top: the sizes of the free subunits (40S and 60S) should be the same as the ones in the following monosome / polysomes.

We have corrected the figures as suggested by the reviewer

2, Fig 1b, Figure legend: polysome instead of “Ploysome”.

We thank the reviewer and we have corrected the typo mistake

3, Fig 1b: For the density of polysome profiling, it is usually categorized into 40S, 60S, monosome (or free subunits and monosome, or prepolyosome), light polysomes, heavy polysomes. It would be better to follow these categories.

We changed the terms accordingly as: free subunits, light polysomes and heavy polysomes

4, Page 5 in the middle- A typo: “In contrast to SL culture, both 2iL and EPI cultures are based on serum-free medium supplemented with small molecule inhibitors”. It should be “molecule”.

We thank the reviewer for pointing out this and we have corrected the typo

5, Fig 1f: in Fig 1d, common vs 2iL shows that there are 28 genes having higher TE in SL/EPI. But in Fig 1f and Fig S3c, there are 30 genes. The author should give some explanations for the discrepancy.

We thank the reviewer for pointing out this discrepancy. Indeed, there are 28 genes in this category and we have corrected the numbers in Fig 1f and Fig S3c accordingly.

6, Page 6: subtitle: “Integrative analysis of ribosome occupancy, RNA expression, and protein expression suggest pervasive translational buffering in ground state pluripotency”. It should be “suggests”.

We thank the reviewer for pointing out this and we have corrected the typo.

7, Fig 3d legend: “Transcriptional: protein change >3 fold and RNA change >2 fold”. It should be “RFP change > 2 fold” based on the text on page 8.

The Transcriptional change refers to genes that change expression at both RNA and protein levels; the original sentence appears to be correct and thus was not changed.

8. Page 2: (DNA>RNA>protein) is better presented as (DNA ↔ RNA → protein)

We have applied the changes as suggested by the reviewer

9. I may have overlooked, but it seems that the data for Fig 3e and 3f were not mentioned/referred to in Results?

We apologize for the negligence in citing these two figures, which are now added in the text.

10. Fig 4e: I found it difficult to understand the right part of the illustration with PD, CH, 2i, CH/PD symbols pointing with --| or black à. May need add some explanations in Fig legends. Also, aren't 2i the same as CH/PD?

We have now removed the symbols and added this description in figure legend for clarity:

“ ‘C/P’ refers to proteins that require either PD or CH to maintain expression similar to 2iL ESCs; only removing both inhibitors from 2iL induces changes resembling the SL/EPI states. ‘2i’ refers to proteins that require both PD and CH signaling to maintain expression similar to 2iL state; removing either PD or CH from 2iL induces changes similar to SL/EPI state. ‘P’ refers to proteins that require PD to maintain an expression pattern similar to 2iL; removing PD from 2iL medium induces changes similar to SL/EPI state. ‘C’ refers to proteins that require CH to maintain an expression pattern similar to 2iL; removing CH from 2iL medium induces changes similar to SL/EPI state.”

Reviewer #2 (Remarks to the Author):

Atlasi and Jafarnejad et al characterized how gene expression changes at different levels during early mouse embryonic stem cell differentiation. They employ a suite of high-throughput approaches including ribosome profiling, proteomics, and RNA sequencing to measure gene expression at the transcriptional, translational, and protein levels. They find that most changes in ultimate gene levels are driven by transcription, and find a buffering role for translational control to maintain protein levels between cell states. It is not clear how cells could achieve buffering at the translational level for an arbitrary gene set that changes in transcription but not protein level, and it makes me wish for attempts to answer this. Is the gene set enriched in particular regulatory elements? How might cells accomplish this? Without this the work ends up being rather descriptive and leaves next steps unclear. The work would be strengthened considerably by testing the hypothesis that translation is buffering transcriptional changes in this system.

We thank the reviewer for the constructive comments that further strengthened this paper.

In this manuscript, we provided the first comprehensive and integrative overview of gene expression program in different states of ESCs and demonstrate the divergence between mRNA-abundance, translation rate and protein expression. As part of this, we also revealed that a number of mRNAs undergo translational buffering to maintain constant protein levels across the different states of ESCs. The phenomenon of translational buffering has been previously observed in several studies and is believed to be part of the robustness of the gene regulatory networks (*MacNeil and Walhout 2011 doi/10.1101/gr.097378.109; Levy and Siegal 2008 DOI:10.1371/journal.pbio.0060264; Masel and Siegal 2009 DOI:10.1016/j.tig.2009.07.005*) in response to genetic and environmental perturbation. However, as this is a complex process that involves multiple layers of gene regulation, tackling the underlying mechanisms has proved a challenging task so far. In this regard, our data suggest a potential insight into such mechanism that involves specific mRNA sequences and differential expression of RBPs that are known regulators of mRNA translation.

We show that the group of mRNAs which undergo translational buffering is not an arbitrary group (**Fig 1e**). Indeed, we identified common structural features shared between these mRNAs. In particular, we show that these mRNAs a) have high percentage of A/U nucleotide composition at their 3' UTR sites b) are highly enriched in AU-rich element in their 3' UTRs and c) have enrichment for binding motifs of the ELAV proteins that specifically recognize the ARE sites. Our analyses thus point to the specific regulation linked to ARE and ELAV proteins in these mRNAs, which are generally believed to enhance mRNA stability and translation efficiency. We are now in the process of addressing this specific regulation in mESCs using different genetic and biochemical approaches. However, we believe deducing the exact mechanism responsible for this phenomenon is beyond the objective of this report and should be addressed in future studies. We would like to emphasize, however, that our data go far beyond identification of mRNAs that are impacted by the translation buffering and will stimulate future studies to address the specified regulation of the various layers of gene expression in ESCs.

Major comments:

1) Throughout the paper the authors make general claims about the behavior of cells as they differentiate. It is unlikely that differentiation between two arbitrary cell types is regulated identically. As such, it is critical that the authors more precisely qualify their claims about differentiation to reflect the differentiation they are studying: ground-to-primed embryonic stem cells.

To make the claim that the 2iL-SL-EpiLSC model is associated with differentiation of ground-to-primed ESCs, we have now compared the expression profile of ESCs and available RNA-seq data from the early mouse embryo (*Boroviak et al, Dev Cell, 2015, doi:10.1016/j.devcel.2015.10.011*). We first compared the profiles of 2iL, SL and EpiLSCs ESCs and selected genes that are specific for each state or shared between SL and EpiLSCs when compared to 2iL-state. We then compared these state-specific genes with available RNA-seq data from developing mouse embryo. As has been reported previously (*reviewed in Atlasi and Stunnenberg Nat Rev Genet 2017, doi:10.1038/nrg.2017.57 and Weinberger et al, Nat Rev Mol Cell Biol 2016, doi:10.1038/nrm.2015.28*), the 2iL-specific genes are mainly enriched in pre-implantation embryo (Morula, ICM and E4.5 Epiblast cells) whereas the EpiLSC-specific genes are mainly enriched in post-implantation E5.5 epiblast. SL-specific genes are enriched in both pre-implantation and post-implantation stage embryo reflecting the 'metastable' state of SL cells that display mixed features of both naïve and primed state pluripotency. Genes that are shared between SL and EpiLSCs are mainly enriched in post-implantation embryo.

Accordingly, 2iL-specific genes (e.g *Bhmt, Dazl* and *Spic*) are highly expressed in pre-implantation embryo and are significantly down-regulated in post-implantation epiblast. Conversely, SL and EPI-specific genes (e.g *Dnmt3b, Otx2, Fgf5* and *Dusp6*) are highly upregulated in post-implantation epiblast. Collectively, these data support the notion that 2iL-ESCs represent a less differentiated ground state pluripotency, whereas the EPI-state represents the primed and post-implantation-like state. SL-ESCs represent an in-between state with features resembling both naïve and primed state pluripotency. We have now added the new analysis as part of **new Supplementary Fig 2** in the manuscript and added the following description in the text:

"Comparing the gene expression profile of 2iL, SL and EPI ESCs and the early developing mouse embryo indicated that 2iL-specific genes are mainly enriched in pre-implantation embryo whereas the EPI-specific genes are mainly enriched in post-implantation epiblast. SL-specific genes were enriched in both pre- and post-implantation stage embryo reflecting the 'metastable' state of SL-ESCs with features resembling both naïve and primed state pluripotency."

2) Translation efficiency (TE) is a noisy metric, as any noise from RNAseq in the denominator is amplified when taking the ratio. The authors also identify a set of genes on page 8 which have a change in RFP of >2 fold but no change in RNA (< 2 fold). Is this set of genes that uniquely change in translation level a better predictor of protein levels than TE?

We analyzed the list of genes in more detail and indeed this list of genes show better correlation between change in RFP and protein level. Some of these genes (26%) were also included in our original list of genes based on differential TE analysis, whereas for the rest, the FDR of TE change did not reach the chosen significance for the cut-off. As here we have selected for genes that show significant change in protein level, the correlation between RFP and protein change is more pronounced. Therefore, we believe that integrating the protein expression data into the ribosome profiling pipeline is instrumental and can significantly improve the identification of genes that undergo specific translational changes.

We have now added the analysis of protein expression for this set of genes as **new supplementary Fig 4e**

3) Please clarify in the methods what the origin of 5' and 3' UTR sequences is: in one section it is stated that they are defined as 50 nt upstream and 200 nt downstream of the CDS, while in another

section it says they were obtained from UCSC BIOMART (which itself is confusing since I normally associate BioMart with Ensembl).

We apologize for the omission in proper citation of the source database. Indeed, we used ENSEMBLE BIOMART database for our analysis and we have now corrected this in the text.

To identify the structural features in different gene sets (including nucleotide composition, sequence enrichment and motif enrichment analysis), we have retrieved the 5'-UTR, CDS and 3'-UTR sequences from BIOMART and used the full-length sequences in this analysis. For Fig S1c and d, our aim was to analyze the sequence around the start and stop codons. This is the common approach used in ribosome profiling analysis; we therefore selected the 50 nt flanking the start site and the 200 nt flanking the stop codon for this analysis. We have now clarified in the text as follows:

“We retrieved the 5'-UTR, CDS and 3'-UTR sequences from ENSEMBLE BIOMART and used the full-length sequences for structural features analysis (including nucleotide composition, sequence enrichment and motif enrichment analysis). To analyze the sequence around the start and stop codons, we selected the 50 nt flanking the start site and the 200 nt flanking the stop codon.”

Minor points:

The authors should indicate the organism they have performed the experiments in in the abstract.
We have now added the organism name in the abstract

The authors may wish to clarify what they are referring to as “82% and 81%” etc on page 6, as from the text it seems they are discussing correlation coefficients and not percents.
We have clarified that we are referring to Pearson’s correlation coefficients in this section and added this in the text

I am not sure that “the translational landscape of early mammalian development ... remains largely unexplored”. The authors may wish to consider replacing this statement with a more precise statement of the gap they seek to fill, as there are hundreds (or maybe thousands) of studies on translational control in early mammalian development. A similar sentence exists on the bottom of page 3. Specifically, the authors should present their results in light of Freimer et al 2018, who studied how translation and mRNA halfives change as mouse ESCs differentiate.
We appreciate the reviewer’s comment on the confusing language regarding the reference to previous studies and we have modified this sentence in the abstract as follows:

“The translational landscape in ground state pluripotency and its impact on cellular proteome, however, is less explored”.

We have also modified the sentence in page 3 of introduction to reflect better the multitude of studies of gene expression program at post-transcriptional and translational levels in ESCs and early mammalian development and the gap that we seek to fill in the current manuscript as follows:

“The transition between different states of pluripotency provides a versatile model for the study of stem cell biology and is of great importance to elucidate early embryonic development. Previous studies investigated the gene expression program at transcriptional and post-transcriptional levels in ESCs and early mammalian development. These observations indicate the importance of machineries that control mRNA translation and decay in ESCs. These studies, however, mainly focused on single aspects of regulation of mRNA stability and/or translational control (e.g. microRNA-induced repression) in conventionally maintained SL-ESC. The concomitant regulation of transcriptome and translome, and its global impact on the cellular proteome remains less understood in ground state 2iL-ESCs. Here we employed several genome-wide approaches ...”

We also discussed our results in light of the study of Fremier, et al. We wish to note, however, that Fremier, et al, employed ESCs cultured in SL+2iL medium. These cells represent a mixed state of serum and 2i-ESCs and, thus are less defined and differ from the ground state pluripotent 2iL-ESCs commonly studied in the field. Nevertheless, the authors observed that transcriptional regulation rather than differential mRNA stability is the dominant regulator of mRNA abundance when ESCs are compared to the committed EpiLSCs. Furthermore, the authors reported differential but limited mRNA translation in the SL+2iL cells when compared to EpiLSCs. Our findings corroborate and extend these observations and suggest that transcriptional control represents the main mode of gene regulation among different states of ESCs whereby specific translational and post-translational regulations play a lesser contribution to regulating the final protein abundance. We added the following text to the discussion section:

“Recently, Fremier et al, investigated the transcriptome-wide changes in mRNA translation and stability in the context of microRNA-induced gene silencing (Fremier et al Elife. 2018). In this regard, microRNAs were shown to regulate both the stability and translation of their target mRNAs. Notably, depletion of DDX6, a RNA helicase implicated in microRNA-induced gene silencing, affected mRNA translation without impacting their stability thus demonstrating a role for translational repression, independent of mRNA destabilization. By measuring nascent mRNA transcription and steady state mRNA levels, Fremier et al also employed ESCs maintained in a mix of SL+2iL culture condition and showed that mRNA transcription rather than change in mRNA stability is the dominant regulator of mRNA abundance during ESCs-to-EpiLSCs differentiation. Furthermore, Fremier et al reported limited differential mRNA translation in the employed ESCs when compared to EpiLSCs. Our findings corroborate and extend these observations and suggest that transcriptional control represent the main mode of gene regulation among different states of ESCs whereby specific translational and post-translational control play a lesser contribution to regulating the final protein abundance.”

AU-rich elements can both increase or decrease translation and mRNA stability (which are also linked). The authors assert on page 11 that ARE's are “generally linked to enhanced stability and translation” and it would be nice if this statement was supported by a reference.

We would like to point out that this particular sentence referred to the stabilizing role of ELAVL proteins on mRNAs and not the AREs; “A potential mechanism for some of these mRNAs could be the activity of the ELAVL; ELAVL1/2/3/4) protein family, which are known to specifically bind the AREs and are generally linked to the enhanced stability and translation of their target mRNAs”. Although we agree that multiple studies also reported that ELAV proteins could decrease mRNA stability or translation efficiency, majority of studies attribute a positive impact on mRNA stability and translation efficiency to these proteins summarized in the following review papers (Hinman et al 2008, doi: 10.1007/s00018-008-8252-6; Simone et al 2013, doi: 10.1016/j.gde.2012.12.006; Brennan et al 2001, doi: 10.1007/PL00000854). However, we appreciate that the language in that sentence is ambiguous and thus we changed it and added the supporting references in order to better reflect the existing literature. We also added a sentence along with relevant references that address the multifaceted role of AREs in regulation of mRNA translation and stability.

“A potential mechanism for some of these mRNAs could be the activity of the ELAVL; ELAVL1/2/3/4) protein family, which are known to specifically bind the AREs. While AREs were demonstrated to both increase or decrease translation and mRNA stability in various contexts (Chen et al 2001, DOI:10.1016/s0092-8674(01)00578-5; Kawai et al 2006, DOI: 10.1128/MCB.26.8.3295-3307.2006; Mazan-Mamczarz et al 2006, DOI:10.1128/MCB.26.7.2716-2727.2006; Brennan et al 2001, doi: 10.1007/PL00000854), ELAV proteins are generally linked to the enhanced stability and translation of their target mRNAs (reviewed in Hinman et al 2008, doi:10.1007/s00018-008-8252-6; Simone et al 2013, doi: 10.1016/j.gde.2012.12.006; Brennan et al 2001, doi: 10.1007/PL00000854). We found a considerable

enrichment of AU-rich elements and the consensus binding motif of the ELAVL proteins in the 3' UTRs of mRNAs that showed higher translation efficiency in 2iL ESCs. Accordingly, we observed a significant downregulation of ELAVL2/4 proteins in both SL and EPI when compared to 2iL state. Thus, downregulation of the ELAVL proteins in SL and EPI cells might be linked to the decreased translation efficiency of the ARE-containing mRNAs in these cells."

REVIEWERS' COMMENTS:

Reviewer #1 (Remarks to the Author):

The authors have addressed all my concerns very well. Especially for the first major point, the authors provided very elaborate explanations with interesting experimental results, related literature, and logical thinking. I think the manuscript is now suitable for publication in NC.

A few very minor typos or points:

1. Page 9: 2iL-SI should be 2iL-SL
2. The authors may want to double check the consistency on "ELAV versus ELAVL"; "EpiLSC versus EpiLC" throughout the manuscript including Figures&Legends and methods.

--Jianlong Wang

Reviewer #2 (Remarks to the Author):

Thanks to the authors for the comprehensive replies to all review comments. In my view the work should be published at this stage.

Two small comments:

I note that there is still a reference to the "UCSC BIOMART" in the final version, which makes me think a final proofreading to catch small mistakes like this might improve the manuscript.

Please deposit and release all associated source code to facilitate interpretation and reproduction of the work (e.g. "peakstats.py"). Thanks.

REVIEWERS' COMMENTS:

Reviewer #1 (Remarks to the Author):

The authors have addressed all my concerns very well. Especially for the first major point, the authors provided very elaborate explanations with interesting experimental results, related literature, and logical thinking. I think the manuscript is now suitable for publication in NC.

A few very minor typos or points:

1. Page 9: 2iL-SI should be 2iL-SL

Applied

2. The authors may want to double check the consistency on "ELAV versus ELAVL"; "EpiLSC versus EpiLC" throughout the manuscript including Figures&Legends and methods.

Applied

--Jianlong Wang

Reviewer #2 (Remarks to the Author):

Thanks to the authors for the comprehensive replies to all review comments. In my view the work should be published at this stage.

Two small comments:

I note that there is still a reference to the "UCSC BIOMART" in the final version, which makes me think a final proofreading to catch small mistakes like this might improve the manuscript.

Applied

Please deposit and release all associated source code to facilitate interpretation and reproduction of the work (e.g. "peakstats.py"). Thanks.

Applied